# Modularized Self-Reflected Video Reasoner for Multimodal LLM with Application to Video Question Answering

**Zihan Song** [1]   **Xin Wang** [1]   **Zi Qian** [2]   **Hong Chen** [1]   **Longtao Huang** [2]   **Hui Xue** [2]   **Wenwu Zhu** [1]

## Abstract

Multimodal Large Language Models (Multimodal LLMs) have shown their strength in Video Question Answering (VideoQA). However, due to the black-box nature of end-to-end training strategies, existing approaches based on Multimodal LLMs suffer from the lack of interpretability for VideoQA: they can neither present reasoning paths nor indicate where the answers are derived from the video. To address this issue, we propose **MSR-ViR** (**M**odularized **S**elf-**R**eflected **Vi**deo **R**easoner), which for the first time integrates modular networks to Multimodal LLMs, capable of providing VideoQA with explicit reasoning paths for more interpretability. Specifically, a **MoST-Grounding** (Modularized Spatial-Temporal Grounding) network is proposed to decompose complex questions via tree-structured policies, localizing relevant temporal and spatial segments within videos through step-by-step reasoning. The proposed MoST-Grounding network provides explicit visually grounded information for Multimodal LLMs with clear reasoning paths, thus enhancing interpretability for the predicted answers. To further improve the reasoning quality, we design an **Alternate Self-reflection Training Strategy** to jointly optimize policy generation and Multimodal LLMs. Experiments on real-world datasets demonstrate the superiority of our proposed MSR-ViR framework in video understanding, reasoning transparency, and providing explicit localization evidence for answers.

[1]Department of Computer Science and Technology, Beijing National Research Center for Information Science and Technology, Tsinghua University, Beijing, China [2]Alibaba Group, Hangzhou, China. Correspondence to: Xin Wang <xin_wang@tsinghua.edu.cn>, Wenwu Zhu <wwzhu@tsinghua.edu.cn>.

*Proceedings of the 42$^{nd}$ International Conference on Machine Learning*, Vancouver, Canada. PMLR 267, 2025. Copyright 2025 by the author(s).

## 1. Introduction

Video Question Answering (VideoQA) is a representative task in video understanding, aiming to answer questions based on the content of a given video. Leveraging their rich external knowledge and strong generalization capabilities, multimodal large language models (Multimodal LLMs) have emerged as powerful tools for tackling video understanding tasks such as VideoQA, video captioning and so on. However, existing approaches based on Multimodal LLMs suffer from the following issue in VideoQA tasks: the classic end-to-end training approaches operate as black-box systems, which inherently suffer from the lack of interpretability. Falling short in terms of transparency, they are unable to unveil the reasoning paths or pinpoint the specific segments of the video from which the answers are derived.

To solve this issue, we marry modular network with Multimodal LLMs for interpretable VideoQA. In particular, we propose the **M**odularized **S**elf-**R**eflected **Vi**deo **R**easoner (MSR-ViR) framework, which is able to obtain clear reasoning paths when answering the questions. The proposed MSR-ViR framework contains a **Mo**dularized **S**patial-**T**emporal **Grounding** (**MoST-Grounding**) module, together with a reinforcement learning-based **Alternate Self-reflection Training strategy** to train a Multimodal LLM for VideoQA, as is shown in Figure 1 (c). Concretely, by following a tree-structured execution policy generated by a Question Parser, MoST-Grounding first decomposes a complex question into several small parts. Then through step-by-step spatial-temporal grounding and reasoning, it progressively derives the most relevant visual information in the video, which will be fed to a Multimodal LLM to answer the question. The execution policy provides a clear reasoning path from the question to the answer, providing interpretability for making predictions, while the spatial-temporal grounding results can provide visual evidence for the predicted answers. As for model optimization, on the one hand, the proposed **Alternate Self-reflection Training strategy** optimizes policy generation via letting the Multimodal LLM undergo Supervised Finetuning (SFT) based on the execution results of the policies generated by the Question Parser in order to better understand the video content. On the other hand, the predicted result of the Multimodal

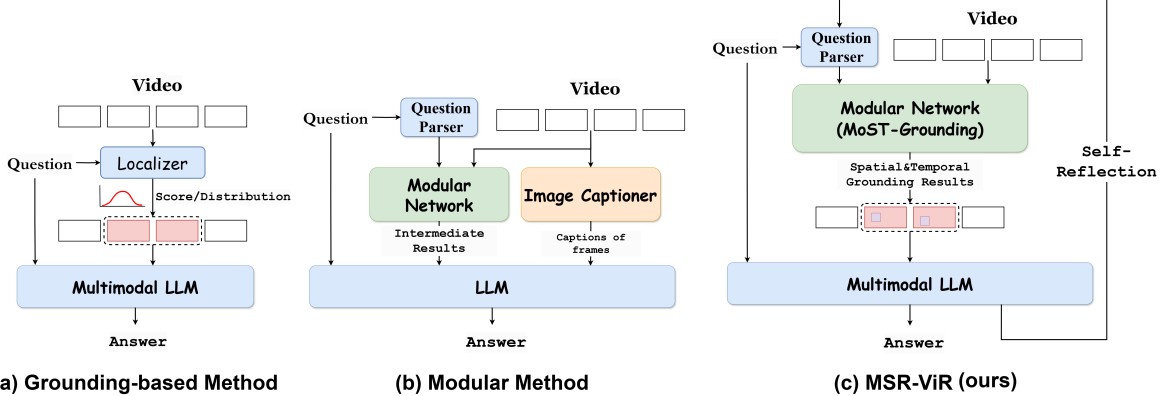

*Figure 1.* Comparison between existing works and our method. (a) shows the common framework of existing grounding-based methods, and (b) shows the common framework of existing modular methods. (c) is our MSR-ViR framework.

LLM also serves as a feedback to train the Question Parser through reinforcement learning. The training of the Question Parser and the Multimodal LLM alternates in a cyclical manner, and both are optimized during this alternate self-reflected training process.

We train the Multimodal LLM with both classic methods and our proposed model over the training sets of commonly used VideoQA datasets, NExT-QA (Xiao et al., 2021) and STAR (Wu et al., 2021), and evaluate them on their corresponding test sets. We also conduct zero-shot experiments on long-form VideoQA datasets EgoSchema (Mangalam et al., 2023) and VideoMME (Fu et al., 2024a) for further evaluation. The results show that our proposed MSR-ViR framework significantly outperforms classic training methods as well as other grounding-based VideoQA methods. Furthermore, we conduct evaluations on NExT-GQA (Xiao et al., 2024), a widely-used grounding-based VideoQA dataset, whose results demonstrate that the proposed MSR-ViR framework is able to not only improve the performance of VideoQA but also more accurately localize the temporal segments relevant to the questions compared to baseline methods. We also prove a theoretical upper bound over the computational complexity of MSR-ViR, demonstrating that the additional computational overhead introduced can be strictly bounded and is reasonable.

Our contributions can be summarized as follows:

- We propose a modularized VideoQA framework with self-reflection training. To the best of our knowledge, this is the first VideoQA work incorporating modular networks into Multimodal LLMs for interpretability.

- We propose a modular network **MoST-Grounding** to decompose complex questions into small parts and derive the most relevant visual information. We also propose an **Alternate Self-reflection Training Strat-**

**egy** to generate reasoning paths.

- We provide a theoretical proof of the upper bound computational complexity of our MSR-ViR framework, demonstrating that the computational overhead is strictly bounded.

- We conduct extensive experiments to demonstrate that i) our MSR-ViR framework can significantly outperform baseline methods, and ii) MSR-ViR can accurately localize the temporal segments, providing visually grounded evidence for the predicted answers.

## 2. Related Works

**Video Understanding with Multimodal LLMs.** Multimodal LLMs have been widely used for video understanding tasks (Zhang et al., 2023; Lin et al., 2024; Maaz et al., 2024; Li et al., 2023b; 2024a; Zhang et al., 2024c; Song et al., 2024; Yao et al., 2024; Li et al., 2025). Most Multimodal LLMs for videos are built on open-source LLMs such as LLaMA (Touvron et al., 2023) and Vicuna (Chiang et al., 2023), and adapters are utilized to align encoded visual information with the textual space. However, classic end-to-end training methods of Multimodal LLMs remain black boxes, resulting in a lack of interpretability as they are unable to provide inference process as well as grounded evidence of the answer in the video.

**Grounded VideoQA with LLMs.** Grounded VideoQA aims to indicate where in the video the answer originates while answering questions. Most existing grounding-based (retrieval-based) VideoQA methods (Wang et al., 2024f; Xiao et al., 2024; Qian et al., 2024b; Yu et al., 2024; Wang et al., 2024a) attempt to localize time segments relevant to the question within the video in the first place and then sample frames from the identified segments to serve as inputs to Multimodal LLMs, as is shown in Figure 1 (a).

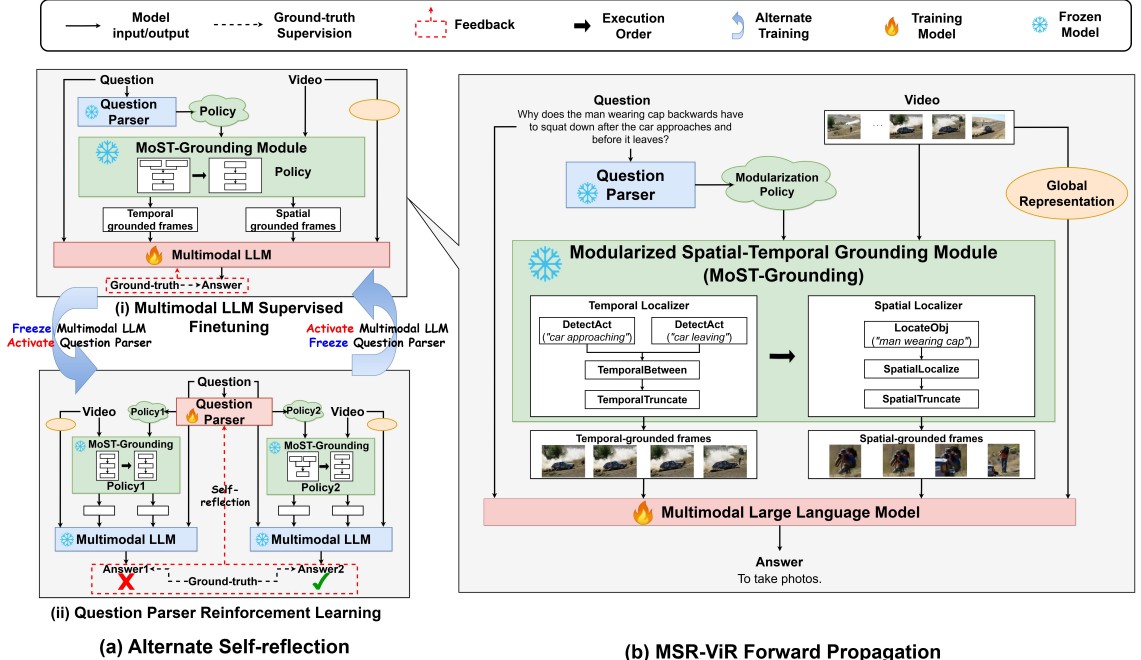

*Figure 2.* Framework of MSR-ViR. The left part (a) shows Alternate Self-reflection Training Strategy, including (i) Multimodal LLM Supervised Finetuning and (ii) Question Parser Reinforcement Learning. The right part (b) demonstrates forward propagation details of MSR-ViR during Multimodal LLM supervised fine-tuning.

Other work like GroundVQA (Di & Xie, 2024) integrates query grounding and answer generation into a unified model to achieve grounded VideoQA. Grounded-based methods address the issue of providing visual evidence in videos to some extent, but still they lack interpretability as they typically rely on black-box models to perform temporal localization without a clear reasoning path, especially for questions with complicated structures.

**Modular VideoQA with LLMs.** Modular methods utilize various smaller models according to execution policies generated by certain LLM to handle sub-tasks derived from the original complex question, and another LLM integrates the outputs of these smaller models to produce the final answer (Min et al., 2024; Zhang et al., 2024a; Surís et al., 2023; Wang et al., 2024c;g), as shown in Figure 1 (b). While this approach enhances interpretability, the unimodal LLMs used can only receive video information through video captions, potentially missing temporal context and detailed information with the video. Additionally, policies generated without training might be unreasonable, affecting accuracy of question answering. Appendix A presents some other works that are related to our work.

## 3. The Proposed MSR-ViR Framework

In this section, we describe our proposed framework MSR-ViR, a modularized VideoQA framework with alternate self-reflection training strategy. Figure 2 demonstrates our overall framework. We will first introduce the Question Parser in Section 3.1, and then introduce our MoST-Grounding module in Section 3.2. In Section 3.3, we will present how to process various information from both MoST-Grounding module and naive inputs based on a Multimodal LLM. Then, our proposed Alternate Self-reflection Training Strategy is introduced in Section 3.4. Finally, the computational complexity of the framework will be discussed in Section 3.5.

### 3.1. Question Parser

Many video language questions actually involve a "multi-step" reasoning process rather than the end-to-end "one-step" processing. As the example shown in Figure 2, our MSR-ViR framework mirrors the "multi-step" reasoning process how humans tackle VideoQA tasks: when facing a video together with a complex question, we first utilize a **Question Parser** to decompose the question into several sub-questions, allowing us to identify the relevant video segments and regions, together with a tree-structured reasoning process, to explicitly help answer the question.

Given a question $q$, our Question Parser $Q$ aims to generate the policy $p = Q(q)$, which serves as the execution plan for the subsequent MoST-Grounding module (illustrated in Section 3.2) to invoke various small modules for temporal and spatial localization. Considering the diversity of ques-

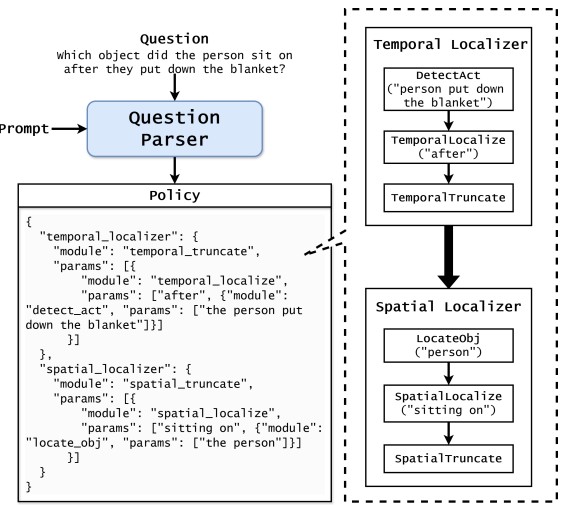

*Figure 3.* Prompting the Question Parser to generate modularization policies.

tion content and structure, as well as the limited training data for question policies, we utilize a large language model as the Question Parser and take advantage of its in-context learning capabilities (Brown et al., 2020). As shown in Figure 3, we carefully design the prompt as an instruction and stimulate the Question Parser to generate policies in the uniform JSON format, organizing and chaining the small modules in a specific structure. The JSON structure allows MoST-Grounding module to recursively call each module, ultimately generating the spatial-temporal grounding results. The complete prompt for the Question Parser is presented in Appendix B.

### 3.2. Modularized Spatial-Temporal Grounding Module

MoST-Grounding module is the core component of our framework, recursively invoking various small modules according to the modular policy generated by the Question Parser to achieve temporal and spatial localization for complex questions. Our MoST-Grounding module consists of two parts: **temporal localizer** $\mathcal{F}_t$ and **spatial localizer** $\mathcal{F}_s$, each containing several small modules for temporal and spatial localization, respectively.

Given a concept $c_t$ and a video $v = \{v_1, v_2, \ldots, v_T\}$ containing $T$ frames, temporal localizer aims to generate the most relevant segments $v_s = \{v_i, \ldots, v_j\}$ from the video $v$, where $1 \leq i \leq j \leq T$. Later the output of the temporal localizer, namely video segments $v_s$, along with concepts $c_s$ are processed to the spatial localizer, generating most relevant visual bounding box $b_{v_s}$ within video segments $v_s$. Formally, MoST-Grounding module $\mathcal{M}$ is written as follows:

$$\mathcal{M}(v, c_t, c_s) = \mathcal{F}_s\big(\mathcal{F}_t(v, c_t), c_s\big). \qquad (1)$$

To address complex semantic scenarios, both the temporal localizer and spatial localizer consist of several types of small modules that can be dynamically assembled according to the policy. To be specific, there are 7 small modules in our MoST-Grounding module. As the core module of temporal localizer, `DetectAct` module temporally localizes a simple action described by a short query (like "car approaching") in the video. In this module we utilize a unified video temporal grounding model UniVTG (Lin et al., 2023). Spatial localizer contains a `LocateObj` module localizing an object described by a short query (like "man wearing cap") in a video frame. In this module we take advantages of an open-vocabulary real-time object detector YOLO-World (Cheng et al., 2024). Details about other modules are presented in Appendix C.

In our execution policies, both temporal localizer and spatial localizer will dynamically assemble corresponding modules $\{m_i^t\}$ and $\{m_i^s\}$, respectively. With the policy $p$ generated by Question Parser, $\mathcal{F}_t$ and $\mathcal{F}_s$ in Equation (1) would be instantiated as follows:

$$\mathcal{F}_{t|p}(\cdot) = \mathcal{I}\big(\{m_i^t\}, p\big)(\cdot), \ \mathcal{F}_{s|p}(\cdot) = \mathcal{I}\big(\{m_i^s\}, p\big)(\cdot). \quad (2)$$

After instantiating with the policy, the modules within the temporal localizer are called first to locate and extract the temporal-grounded frames from the video. Subsequently, the modules within the spatial localizer are invoked to generate the corresponding spatial-grounded frame for each temporal-grounded frame. In this modular manner, MoST-Grounding extracts several temporally and spatially localized video frames from the video, which will serve as visual input to our Multimodal LLM.

### 3.3. Multimodal LLM Answerer

After MoST-Grounding localizes the temporal segments and spatial regions relevant to the question, a Multimodal LLM is needed to understand the textual and visual information in order to answer the question. Formally, the answer of a question $q$ given the video $v$ can be written as follows:

$$\hat{y}(q, v) = \mathcal{F}\big(q, v_s, b_{v_s}\big), \qquad (3)$$

where $v_s$ and $b_{v_s}$ represent the video segments and bounding box generated from MoST-Grounding module, and $\mathcal{F}(\ldots)$ denotes the forward propagation of Multimodal LLM. To better enhance the video understanding ability of the Multimodal LLM, we extend the input of the Multimodal LLM in Equation (3) with the following two strategies. Firstly, we provide an additional global representation of the video to the Multimodal LLM by compressing several uniformly sampled frames through average pooling. This is necessary because the MoST-Grounding module may not always accurately localize the segments relevant to the question, and the Multimodal LLM might overlook essential information if it

relies solely on the grounding results. Secondly, in addition to the original question and the aforementioned visual information, we provide the Multimodal LLM with a guiding prompt that explains the specific meanings of various visual input components. With these two designs, Equation (3) would be modified with:

$$\hat{y}(q, v) = \mathcal{F}\big(Pt(q, v_s, b_{v_s}, g_v)\big), \qquad (4)$$

where $Pt(\dots)$ represents the guiding prompt and $g_v$ denotes the global video representation. Specific input format of the Multimodal LLM is presented in Appendix F.

With the ground-truth of question $q$ being $y$, the supervised finetuning loss of the Multimodal LLM is defined as a cross entropy loss $\mathcal{L}_{\text{CE}}$:

$$\mathcal{L}_{\text{CE}}(\hat{y}(q, v), y) = - \sum_{(v,q,y) \in D} y \log(\hat{y}(q, v)), \qquad (5)$$

where $y$ is the target answer, $D$ is the dataset. By optimizing the loss function in Equation (5), the Multimodal LLM undergoes supervised finetuning on VideoQA datasets, learning to answer questions based on all the provided information.

### 3.4. Alternate Self-reflection Training Strategy

As discussed in Section 3.1, we can teach the Question Parser to generate modularization policies from complex questions by providing examples in the prompt. However, relying solely on in-context learning does not ensure the quality of the policies. To address this issue, we propose the Alternate Self-reflection Training Strategy, which enables the Question Parser to improve the quality of its policies through reinforcement learning.

We assume that for a given question, provided that all our modules remain unchanged, a reasonable modular policy is more likely to accurately localize the correct temporal segments and spatial regions. Consequently, the loss computed by the Multimodal LLM is likely to be smaller. Therefore, we provide feedback to the Question Parser using the loss noticed during the training process of the Multimodal LLM, thereby guiding it through reinforcement learning training. Specifically, we utilize Direct Preference Optimization (DPO) (Rafailov et al., 2024) to train the Question Parser. Different from previous RLHF methods, DPO directly optimizes a language model without explicit reward models, making the training process straightforward and stable. Specifically in DPO, the policy objective is formulated as:

$$\mathcal{L}_{\text{DPO}}(\pi_\theta; \pi_{\text{ref}}) = -\mathbb{E}\left[\log \sigma\left(\hat{r}_\theta(p_w, q) - \hat{r}_\theta(p_l, q)\right)\right], \quad (6)$$

where $\hat{r}_\theta(x, y) = \beta \log \frac{\pi_\theta(x|y)}{\pi_{\text{ref}}(x|y)}$ is the reward of policy $x$ for question $y$ implicitly defined by the language model $\pi_\theta$

and reference model $\pi_{\text{ref}}$, $p_w$ denotes the positive policy, $p_l$ denotes the negative policy and $q$ is the input question. $\pi_\theta$ is our Question Parser to be trained, while $\pi_{\text{ref}}$ is a reference model initialized with the Question Parser itself but remain frozen. Through DPO training, the probability of generating positive policies increases, while the probability of generating negative policies decreases. In other words, the Question Parser learns to generate more reasonable policies. We prompt the Question Parser to view the same question from multiple perspectives, generating different modular policies. The MoST-Grounding module executes each policy, producing their respective grounding results. The Multimodal LLM then computes the corresponding losses. We classify the policy with the smaller loss as positive and the one with the larger loss as negative, training the Question Parser according to Equation (6).

Our training strategy alternates between SFT of the Multimodal LLM and reinforcement learning for the Question Parser, optimizing with the loss functions in Equation (5) and Equation (6), respectively. While training one large model, the other model's parameters remain frozen. During this process, the Multimodal LLM periodically pauses to adapt based on the modular policies from the Question Parser. After a set of training period, the Question Parser utilizes these refined policies to further train the Multimodal LLM, allowing both models to optimize continuously. See Appendix D for detailed training process.

### 3.5. Computational Complexity

Considering that our framework introduces an LLM as the Question Parser and several smaller grounding models as sub-modules in MoST-Grounding, we provide an analysis of the upper bound computational complexity of our framework. We present the following propositions:

**Proposition 3.1.** *Given parameters of Multimodal LLM $P_1$, $P_2$, $P_3$, $P_4$, video with $N$ input frames and resolution $H \times W$, text input with length $l$, the complexity of Multimodal Answerer is $O\left(P_1 N H^2 W^2 + P_2 N^2 + P_3 l^2 + P_4 N l\right)$.*

**Proposition 3.2.** *Given small modules in MoST-Grounding with complexity $c_1$, $c_2$, and large models in Question Parser and Multimodal LLM with complexity $C_1$, $C_2$, $C_3$, question with length $L$, prompt for Question Parser with length $l_p$, video with length $T$ and resolution $H \times W$, the complexity of MSR-ViR is $O\left(c_1 T H^2 W^2 + c_2 T^2 + C_1 H^2 W^2 + C_2 L^2 + C_3 l_p^2\right)$.*

Detailed proof of these propositions is provided in Appendix H. It is worth noting that $c_1$, $c_2$ are significantly smaller than $C_1$, $C_2$, $C_3$, and $N$ is a small constant in our framework. Therefore, compared to its baseline Multimodal LLM, the additional complexity introduced by our framework mainly stems from $C_3 l_p^2$, which is the complexity of Question Parser. This computational overhead is bounded

*Table 1.* Experiments on **NExT-QA** and **STAR-sub**. All models are finetuned on the corresponding training set.The first part contains small vision-language models, and in the second part models or methods are based on Multimodal LLMs. Qwen-VL and LLaVA-NeXT are our direct baselines, MSR-ViR$_Q$ is our framework based on Qwen-VL and MSR-ViR$_L$ is our framework based on LLaVA-NeXT. **Bold** number denotes the best result. Model size and inference speed could be found in Appendix G.2.

| Method | NExT-QA | | | | STAR-sub | | |
|---|---|---|---|---|---|---|---|
| | Temporal | Causal | Descriptive | Average | Interaction | Sequence | Average |
| ATP | 49.3 | 48.6 | 65.0 | 51.5 | 50.6 | 52.8 | 51.7 |
| MIST | 56.6 | 54.6 | 66.9 | 57.1 | 55.5 | 54.2 | 54.9 |
| CoVGT | 57.4 | 58.8 | 69.3 | 60.0 | - | - | - |
| HiTeA | 58.3 | 62.4 | 75.6 | 63.1 | - | - | - |
| InternVideo | 58.5 | 62.5 | 75.8 | 63.2 | 62.7 | 65.6 | 64.4 |
| BLIP-2 | 64.9 | 69.7 | 79.4 | 69.6 | 65.4 | 69.0 | 67.2 |
| TGB | 66.5 | 72.8 | 81.2 | 72.1 | - | - | - |
| InstructBLIP | 70.5 | 71.5 | 79.8 | 72.5 | - | - | - |
| SeViLa | 69.4 | 74.2 | 81.3 | 73.8 | 63.7 | 70.4 | 67.1 |
| GCG | **72.6** | 74.2 | 80.7 | 74.6 | - | - | - |
| Qwen-VL | 68.4 | 71.3 | 80.6 | 71.9 | 60.4 | 65.5 | 63.0 |
| LLaVA-NeXT | 69.5 | 73.3 | 79.7 | 73.1 | 67.6 | 72.1 | 69.9 |
| MSR-ViR$_Q$(ours) | 69.9 | 73.4 | **81.5** | 73.6 | 64.8 | 68.0 | 66.4 |
| MSR-ViR$_L$(ours) | 72.2 | **74.6** | 80.9 | **74.9** | **68.9** | **73.1** | **71.0** |

because Question Parser utilizes prompt with fixed length $l_p$ irrelevant to the complexity of VideoQA scenario. It is also reasonable, as Question Parser generates modular policies as clear reasoning paths enhancing interpretability of our framework. We also conduct inference speed experiments on NExT-QA dataset, and the results are presented in Appendix G.2.

## 4. Experiments

In this section, we first introduce the basic setups of our experiments in Section 4.1. Next, we introduce our experiments on VideoQA datasets NExT-QA, STAR in Section 4.2, after which experiments on long-form VideoQA datasets EgoSchema and VideoMME will be discussed in Section 4.3. Then in Section 4.4 we present our experiments on grounded VideoQA dataset NExT-GQA. Finally, we present the ablation study in Section 4.5.

### 4.1. Experiments Setups

**Datasets.** We conduct experiments on VideoQA datasets NExT-QA (Xiao et al., 2021), STAR (Wu et al., 2021) and long-form VideoQA datasets EgoSchema (Mangalam et al., 2023) and VideoMME (Fu et al., 2024a), together with a grounded VideoQA dataset NExT-GQA (Xiao et al., 2024). NExT-QA and STAR contain various reasoning tasks including spatial-temporal reasoning, logical attribution and so on, while EgoSchema and VideoMME include longer videos and more complex questions, making them particularly well-suited for evaluating video understanding and reasoning capabilities of models. NExT-GQA is derived

from NExT-QA, providing ground-truth temporal clips for validation and test sets to evaluate temporal grounding accuracy. It is worth mentioning that for STAR, we create a subset (STAR-sub) with **Interaction** and **Sequence** questions (82.5% of STAR), excluding **Prediction** and **Feasibility** types as they lack temporal and spatial grounding in videos, making them unsuitable for our framework.

**Baselines.** On NExT-QA and STAR, our baselines include vision-language models ATP (Buch et al., 2022), MIST (Gao et al., 2023), CoVGT (Xiao et al., 2023), HiTeA (Ye et al., 2023), InternVideo (Wang et al., 2022), Multimodal LLMs BLIP2 (Li et al., 2023a), InstructBLIP (Dai et al., 2023) and grounding-based Multimodal LLMs TGB (Wang et al., 2024f), SeViLa (Yu et al., 2024), GCG (Wang et al., 2024a). On EgoSchema and NExT-GQA, our baselines include vision-language models VGT (Xiao et al., 2022), VIOLETv2 (Fu et al., 2023), Temp[CLIP] (Radford et al., 2021), FrozenBiLM (Yang et al., 2022) (which achieve grounded VideoQA with the method in (Xiao et al., 2024)), grounding-based method TGB, SeViLa, LangRepo (Kahatapitiya et al., 2025) and modular method LLoVi (Zhang et al., 2024a), MoReVQA (Min et al., 2024). On VideoMME, our baselines include Video LLMs Video-LLaVA (Lin et al., 2024), ShareGPT4Video (Chen et al., 2024), LongVA (Zhang et al., 2024b), Video-CCAM (Fei et al., 2024b) and VITA 1.5 (Fu et al., 2024b). We utilize Qwen-VL (Bai et al., 2023) and LLaVA-NeXT (Zhang et al., 2024c) as direct baselines on NExT-QA, STAR and EgoSchema, while Qwen2-VL (Wang et al., 2024b) and LLaVA-Video (Zhang et al., 2024e) are our direct baselines on VideoMME.

**Implementations.** We implement our method based on

*Table 2.* Zero-shot experiments on **EgoSchema**. **Sub.** shows the results on EgoSchema subset and **Full** shows the results on EgoSchema full set. The numbers in parentheses represent the improvement of our method compared to their direct baselines. Bold number denotes the best result.

| Method | Size | Sub. | Full |
|---|---|---|---|
| SeViLa | 3B | 25.7 | 22.7 |
| LLoVi | 7B | 50.8 | 33.5 |
| LangRepo | 7B | 60.8 | 38.9 |
| Qwen-VL | 7B | 44.6 | 36.8 |
| MSR-ViR$_Q$(ours) | 7B | 49.0(+4.4) | 38.7(+1.9) |
| LLaVA-NeXT | 7B | 54.8 | 43.3 |
| MSR-ViR$_L$(ours) | 7B | **61.2(+6.4)** | **46.0(+2.7)** |

*Table 3.* Zero-shot experiments on **VideoMME**. **Short**, **Medium**, **Long** represents 3 subsets with different video length. The numbers in parentheses represent the improvement of our method compared to their direct baselines. Bold number denotes the best result.

| Method | Short | Medium | Long | Avg. |
|---|---|---|---|---|
| Video-LLaVA | 45.3 | 38.0 | 36.2 | 39.9 |
| ShareGPT4Video | 48.3 | 36.3 | 35.0 | 39.9 |
| LongVA | 61.9 | 51.4 | 45.4 | 52.9 |
| Video-CCAM | 62.2 | 50.6 | 46.7 | 53.2 |
| VITA 1.5 | 67.0 | 54.2 | 47.1 | 56.1 |
| Qwen2-VL | 65.2 | 52.2 | 48.3 | 55.3 |
| MSR-ViR$_{Q2}$(ours) | 66.8 | 55.4 | 51.3 | 57.9(+2.6) |
| LLaVA-Video | 69.7 | 56.6 | 49.3 | 58.5 |
| MSR-ViR$_{LV}$(ours) | **72.3** | **60.7** | **52.6** | **61.9(+3.4)** |

SWIFT framework (Zhao et al., 2025) and utilize LoRA (Hu et al., 2022) during supervised finetuning. We utilize a large language model Qwen2-7B (Yang et al., 2024) for our Question Parser. Qwen-VL (Bai et al., 2023), LLaVA-NeXT (Zhang et al., 2024c), Qwen2-VL (Wang et al., 2024b) and LLaVA-Video (Zhang et al., 2024e) are utilized for our Multimodal LLMs, denoted as MSR-ViR$_Q$, MSR-ViR$_L$, MSR-ViR$_{Q2}$ and MSR-ViR$_{LV}$ respectively. Following the classic training strategy, we uniformly sample 4 frames from videos for Qwen-VL, 8 frames for LLaVA-NeXT, 32 frames for Qwen2-VL and LLaVA-Video to implement our direct baselines. For MSR-ViR$_Q$, we sample 2 temporal- and 2 spatial-grounded frames. For MSR-ViR$_L$, we sample 8 temporal- 8 spatial-grounded frames. For MSR-ViR$_{Q2}$ and MSR-ViR$_{LV}$, we sample 16 temporal- and 16 spatial-grounded frames. As for our Alternate Self-reflection Training Strategy, the period for alternating training between two LLMs is 200 steps, with the gradient accumulation step set to 16. We conduct 5 epochs of SFT on NExT-QA and STAR for Qwen-VL, LLaVA-NeXT and our MSR-ViR framework, selecting the best model according to the results on validation set. On EgoSchema, we present zero-shot results. We conduct 1 epoch of SFT on a subset of LLaVA-Video-178K (Zhang et al., 2024e) for Qwen2-VL, LLaVA-Video and our corresponding MSR-ViR framework, and present zero-shot results on VideoMME.

## 4.2. Experiments on VideoQA

We compare our MSR-ViR framework with existing vision-language models, Multimodal LLMs and grounding-based methods on NExT-QA and STAR-sub. As shown in Table 1, MSR-ViR$_L$ achieves best results on the overall NExT-QA and STAR-sub dataset together with most subsets formed by different types of questions. Particularly, MSR-ViR$_L$ surpasses TGB and SeViLa, which also utilize grounding-based Multimodal LLMs, on **Temporal** questions where temporal information is essential, demonstrating the superior tempo-

ral understanding ability of our method comparing to previous grounding-based methods. Besides, for **Interaction** questions where spatial information is relatively important, our method MSR-ViR$_L$ also presents the best performance. Comparing MSR-ViR$_Q$ and MSR-ViR$_L$ with their own direct baselines, we prove that our framework help enhance VideoQA abilities of Multimodal LLMs by providing them with most relevant grounded information, ignoring redundant information that may impair understanding.

## 4.3. Experiments on Long-form VideoQA

To further evaluate the performance of our method in complex scenarios, we conduct zero-shot experiments on a long-form VideoQA dataset EgoSchema (Mangalam et al., 2023). The long videos and complex questions in EgoSchema makes it an ideal benchmark for testing a model's ability to accomplish intricate VideoQA tasks. We selected several methods with comparable LLM sizes to ours as baselines, with the results shown in Table 2. **Size** refers to the number of parameters in the largest LLM utilized in each method. As shown, our method MSR-ViR$_L$ achieves the best results on both the EgoSchema subset and the full set, demonstrating its superiority. Additionally, compared to their respective direct baselines, both MSR-ViR$_Q$ and MSR-ViR$_L$ show significant improvements, indicating that our method enhances the ability of Multimodal LLMs to understand long videos and complex questions.

To further demonstrate the superiority of our method, we also conduct experiments on a longer VideoQA dataset VideoMME (Fu et al., 2024a), which contains hour-long videos. Limited by total frame number, Qwen-VL and LLaVA-Next are not suitable for this dataset, so we utilize Qwen2-VL and LLaVA-Video, and the results are demonstrated in Table 3. As shown, both MSR-ViR$_{Q2}$ and MSR-ViR$_{LV}$ significantly outperform the relative baseline. It could

*Table 4.* Experiments on **NExT-GQA**. VGT, VIOLETv2, Frozen-BiLM, Temp[CLIP], TGB, SeViLa, MSR-ViR$_Q$(ours) and MSR-ViR$_L$(ours) are finetuned on the training set. Method in gray lines utilize significantly larger LLMs (Palm-2 and GPT-4). **Bold** number denotes the best result excluding gray methods.

| Method | mIoU | IoU @0.3 | IoU @0.5 | Acc @GQA |
|---|---|---|---|---|
| VGT | 3.0 | 3.6 | 1.7 | 14.4 |
| VIOLETv2 | 3.1 | 4.3 | 1.3 | 12.8 |
| Temp[CLIP] | 12.1 | 17.5 | 8.9 | 16.0 |
| FrozenBiLM | 9.6 | 13.5 | 6.1 | 17.5 |
| TGB | 19.9 | 23.3 | 11.2 | - |
| LangRepo | 8.7 | - | 6.0 | 11.2 |
| SeViLa | 21.7 | 29.2 | 13.8 | 16.6 |
| LLoVi(Mistral-7B) | 8.7 | - | 6.0 | 11.2 |
| LLoVi(GPT-4) | 20.0 | - | 15.3 | 24.3 |
| MoReVQA(Palm-2) | 19.7 | - | 15.4 | 39.6 |
| MSR-ViR$_Q$(ours) | 22.8 | 33.0 | **16.4** | 18.5 |
| MSR-ViR$_L$(ours) | **23.4** | **33.6** | **16.4** | **18.6** |

*Table 5.* Ablation study on NExT-QA and NExT-GQA. **Bold** number denotes the best result. SR means self-reflection, RP means reasoning path, prompts means instruction prompts, $m^s$ represents spatial modules and $g_v$ means global representation of the video.

| | NExT-QA | | | |
|---|---|---|---|---|
| | Tem. | Cau. | Des. | Avg. |
| MSR-ViR$_Q$ | **69.9** | **73.4** | 81.5 | **73.6** |
| w/o SR | 67.2 | 72.5 | 80.5 | 72.1 |
| w/o $m^s$ | 67.0 | 72.5 | 81.4 | 72.2 |
| w/o prompts | 68.3 | 72.4 | **82.4** | 72.8 |
| w/o $g_v$ | 66.9 | 70.1 | 78.0 | 70.4 |
| only w/ $g_v$ | 63.3 | 68.4 | 77.5 | 68.3 |

| | NExT-GQA | | | |
|---|---|---|---|---|
| | Acc @QA | Acc @GQA | mIoU | IoU @0.5 |
| MSR-ViR$_Q$ | **69.9** | **18.5** | **22.8** | **16.4** |
| w/o SR | 68.3 | 17.9 | 22.2 | 15.7 |
| w/o RP | 66.8 | 14.4 | 18.5 | 11.4 |

also be seen that the improvement is more significant on longer subsets, where grounding and reasoning are more essential for VideoQA, further demonstrating the superiority of our MSR-ViR framework.

### 4.4. Experiments on Grounded VideoQA

To further confirm MSR-ViR is capable of more accurately grounding the relevant information thus enhancing VideoQA ability of Multimodal LLMs, we conduct experiments on the NExT-GQA (Xiao et al., 2024) dataset. NExT-GQA not only contains the answer to the question, but also presents a human-annotated ground-truth time span, indicating where the answer is derived from the video, in other words, the most relevant time period to the question. The dataset requires VideoQA models to provide "evidence" of their answer, evaluating the grounding accuracy with **IoP** (Intersection over Prediction) and **IoU** (Intersection over Union). It also measures **Acc@GQA**, which is the proportion of questions that are correctly answered, and at the same time **IoP** between predicted time span and ground-truth time span is larger than 0.5.

We compare our MSR-ViR framework with existing grounding-based methods and modular methods, together with vision-language models and Multimodal LLMs which utilizes **NG+** method in (Xiao et al., 2024) for training. The results are shown in Table 4. Here we only present the results of **IoU** and **Acc@GQA** in Table 4, and the complete results can be found in Appendix G.1. Methods in the first part are models implemented with **NG+**, while the second part includes grounding-based methods and the third part contains modular methods. We de-emphasize meth-

ods implemented with significantly larger LLMs (LLoVi with GPT-4 for example) for fair comparison. MSR-ViR$_Q$ achieves best results on mIoP, while MSR-ViR$_L$ achieves best results on mIoU, indicating that our method grounds the temporal segment relevant to the question more precisely than existing grounding-based and modular methods. The best result of Acc@GQA demonstrates that our method can perform VideoQA tasks more effectively, while also providing more reasonable temporal evidence indicating which specific segment of the video the answer derives from.

### 4.5. Ablation Study

As demonstrated in Table 5, to further validate the effectiveness of modules and designs in our MSR-ViR framework, we conduct ablation study on NExT-QA and NExT-GQA dataset for MSR-ViR$_Q$ concerning the following questions:

**Is Alternate Self-reflection Training Strategy necessary?** We remove the self-reflection training process, only finetuning our Multimodal LLM without training the Question Parser, and the results are shown by w/o self-reflection in Table 5. The average accuracy on NExT-QA declines by 1.5, and accuracy on each subsets decreases to varying degrees. The grounded accuracy as well as IoU of temporal grounding also decline as shown in experiments on NExT-GQA. This demonstrates the necessity of Alternate Self-reflection Training Strategy. To further verify the necessity of reasoning path, we utilize MoST-Grounding to directly provide grounding results based on original questions without reasoning paths, shown in Table 5 as "w/o RP".

**Is spatial_localizer necessary in MoST-Grounding module?** Most existing grounding-based methods only consider

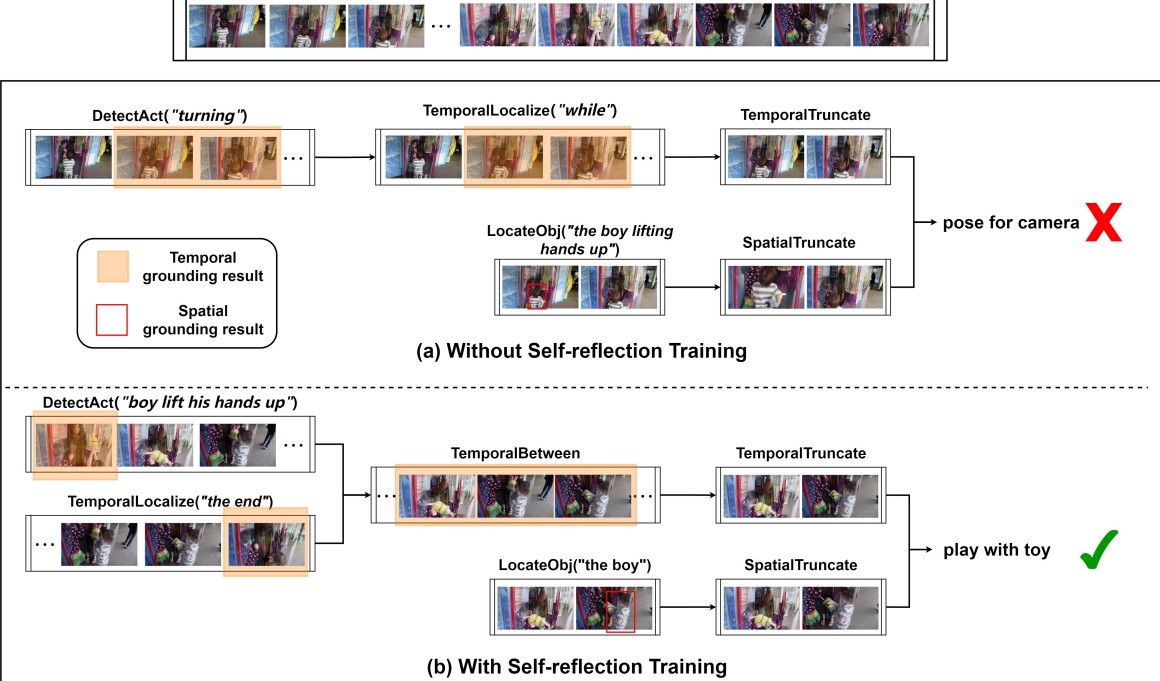

*Figure 4.* An example of MSR-ViR reasoning process. (a) is the reasoning process of MSR-ViR without self-reflection training. (b) is the reasoning process of MSR-ViR with self-reflection training.

temporal grounding, so we remove the **spatial_localizer** including all small modules in it, only providing our Multimodal LLM with temporal grounding results, denoted by w/o spatial modules. The average accuracy on NExT-QA drops by 1.4, proving that spatial grounding results provided by **spatial_localizer** contain useful information for Multimodal LLM to answer the question correctly.

**Are our designs in training Multimodal LLM necessary?** In Section 3.3, we introduce two designs for our Multimodal LLM training: global representation and instruction prompts. We remove these two designs separately and conduct tests on NExT-QA. The results show that the average accuracy on NExT-QA decreases to varying degrees for both, indicating that the two designs we proposed for training Multimodal LLMs are effective. We also attempt to make Multimodal LLMs answer questions utilizing only the information from global representation, results of which are shown in Table 5 as "only w/ $g_v$".

In addition to the above questions, to investigate the choice of small modules in MoST-Grounding, we replace UniVTG in MoST-Grounding with different grounding models and conduct experiments on the NExT-GQA dataset, the results of which are presented in Appendix G.3. We also conduct ablation study on frame sampling strategy, and the results

are presented in Appendix G.4.

Figure 4 demonstrates the reasoning process of MSR-ViR with and without self-reflection training. As shown in Figure 4, after self-reflection training, Question Parser generates more reasonable policies accurately grounding the questions in the videos, leading to correct answers. More visualization examples can be found in Appendix E.

## 5. Conclusion

In summary, we propose Modularized Self-Reflected Video Reasoner (MSR-ViR), a self-reflected framework that integrates a Modularized Spatial-Temporal Grounding (MoST-Grounding) module into a Multimodal LLM for interpretable VideoQA. Modularization policies generated by a Question Parser demonstrates clear reasoning paths enhancing interpretability of our framework, while spatial-temporal grounding results present visual evidence for answers. Through the proposed alternate self-refection training process, policies are gradually refined, becoming more reasonable. Extensive experiments demonstrate that MSR-ViR significantly improves VideoQA capabilities of Multimodal LLMs while grounding answers in videos more accurately. Future work could explore further enhancements to the design of modular network and its execution efficiency.

## Acknowledgements

This work is supported by National Natural Science Foundation of China No.62222209, Beijing National Research Center for Information Science and Technology under Grant No.BNR2023TD03006.

## Impact Statement

This paper presents work whose goal is to advance the field of Machine Learning. There are many potential societal consequences of our work, none of which we feel must be specifically highlighted here.

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

## A. Other Related Works

**Video Grounding with Multimodal LLM.** In order to explore whether Multimodal LLMs are capable of understanding the temporal information in videos, some works have attempted to study the temporal perception ability of Multimodal LLMs for tasks like video grounding (Lan et al., 2023). For example, VTimeLLM (Huang et al., 2024) introduces a boundary perception training process on multi-event datasets, improving the performance of Multimodal LLMs on tasks such as video grounding and dense video captioning. HawkEye (Wang et al., 2024d) constructs a large-scale video-text corpus with segment-level captions and negative spans, on which a coarse-grained segment representation method and a recursive grounding strategy are utilized to train Multimodal LLMs. Other works (Ren et al., 2024; Qian et al., 2024a; Li et al., 2024b; Chen et al., 2023; Feng et al., 2023) also explore the grounding ability of Multimodal LLMs.

**Visual/Video Question Answering with Modular Method.** The modular approach (Wang et al., 2025), which decomposes complex questions into multiple sub-questions and solves each of them through dedicated models, has been attempted in visual/video question answering tasks (Qian et al., 2022; Duan et al., 2022; Nie et al., 2024; Zhang et al., 2024d; Zhong et al., 2024; Khademi et al., 2023; Wang et al., 2024e; Fan et al., 2024; Choudhury et al., 2024; Shi et al., 2024; Ayyubi et al., 2025; Fei et al., 2024a). Specifically, MM-Reasoner (Khademi et al., 2023) leverages vision APIs and LLMs to extract and utilize query-specific knowledge for visual question answering. (Zhong et al., 2024) introduces Visual Table, a novel visual representation that provides detailed object descriptions and knowledge in structured text, significantly boosting performance in visual reasoning tasks. ProViQ (Choudhury et al., 2024) and ENTER (Ayyubi et al., 2025) utilize LLM to generate executable programs, leveraging external tools for modular video question answering. VideoAgent (Fan et al., 2024) and MotionEpic (Fei et al., 2024a) utilize Chain-of-Thought which involves multi-round conversation with LLM leveraging several expert small models, while AoTD distills knowledge from CoT into Video LLMs to improve instruction tuning. Different from existing methods, MSR-ViR is the first framework that not only integrates a modular network into a Multimodal LLM but also jointly optimizes them with self-reflection training for reasoning path refinement and QA accuracy improvement.

## B. Prompt for Question Parser

We carefully design a prompt for our Question Parser to generate policies from given questions from in-context learning. The complete prompt is presented in Figure 5, Figure 6, Figure 7. We first inform the Question Parser of some basic information and an introduction to the functions of each module. Then we tell it the general template of the policy and several variants under special circumstances. Finally, through a few examples, we teach the Question Parser how to generate a policy based on the question.

## C. Module Implementation

*Table 6.* Illustration of our modules. **Type** is the type of the module, where T represents modules in **temporal_localizer** and S represents modules in **spatial_localizer**. **Vision-Language Model** denotes the small vision-language model we use in the module. As for the **input** and **output**, $V$ denotes the input video, $I$ denotes the video frame, $TS$ denotes a time period, $BBOX$ denotes a bounding box, $Q_a$ denotes the action query, $Q_o$ denotes the object query, $p$ is a preposition representing certain temporal relationship and $r$ is a word or phrase representing certain spatial relationship.

| Module Name | Type | Vision-Language Model | Input | Ouput |
|---|---|---|---|---|
| DetectAct | T | UniVTG | $V, Q_a$ | $TS$ |
| TemporalLocalize | T | - | $p, TS^{(in)}$ | $TS^{(out)}$ |
| TemporalBetween | T | - | $TS_1, TS_2$ | $TS^{(out)}$ |
| TemporalTruncate | T | - | $V, TS$ | $I_1, I_2, ...$ |
| LocateObj | S | YOLO-World | $I, Q_o$ | $BBOX$ |
| SpatialLocalize | S | YOLO-World | $I, BBOX^{(in)}, r$ | $BBOX^{(out)}$ |
| SpatialTruncate | S | - | $I, BBOX_1, BBOX_2, ...$ | $I^{(out)}$ |

All small modules in our MoST-Grounding module are listed in Table 6, the detailed implementation of which are as follows:

You are a question parser. You can decompose complex questions into smaller ones, and by doing so it is easier to answer the question. The question is given together with a video clip, and now I hope to temporally and spatially ground the question in the video clip before answering it. The question parser need to decompose the question into several modules, and by solving each module step by step, the question can be grounded in the video. Now I have and only have these modules:

1. detect_act(video_path, act_query): This module works as a temporal localizer. Provided with the path of a video and a query of an action, the module will temporally locate the query into the video, and return a timespan: [start_time, end_time];

2. temporal_localize(prep, act_time): This module locate a timespan in the video. act_time is usually the return value of detect_act, which looks like [start_time, end_time], and prep is a preposition ("before", "when", "after" or something like that). The module determine the final timespan according to the preposition and the act_time, also returning in the format of [start_time, end_time];

3. temporal_between(act_time1, act_time2): This module is actually a special form of temporal_localize, where the preposition is "between". So act_time1 and act_time2 are two timespans detected by "detect_act" module, and this module locate the timespan between the two actions, resulting in returning [start_time, end_time] as well.

4. temporal_truncate(video_path, tp_result): This module truncate a clip of video according to the provided timespan tp_result: [start_time, end_time]. The module samples 2 frames between the start_time of the video and the end_time of the video, thus returning a list of frames: [frame1, frame2];

5. locate_obj(image, obj_query): This module works as a spatial localizer, which spatially locate an area in the image according to the provided obj_query, returning a bounding box of the area: [x1, y1, x2, y2].

6. spatial_localize(image, rel, sp_result): This module locate a specific area in an image. sp_result is the return value of module "locate_obj" which is a bounding box, and rel is a verb or preposition that describes the relationship between the expected result and the given sp_result. The return of this module is the expected result that is also a bounding box.

7. spatial_truncate(image, sp_result): This module truncate the bounding box sp_result in the provided image, resize the bounding box area to the same size of the original image, and returning the resized image.

Among all these modules, 1, 2, 3, 4 are temporal modules and 5, 6, 7 are spatial modules. What you need to do is to provide a policy according to the given question using the above 7 modules. A policy always has two parts: temporal_localizer and spatial_localizer, each utilizing one or some of the above 7 modules.

Generally speaking, a reasonable policy will be in a format as follows:

```
{
    "temporal_localizer": {
        "module": "temporal_truncate",
        "params": [{
            "module": "temporal_localize",
            "params": [$(a preposition), {"module": "detect_act", "params": [$(description of an event)]}]
        }]
    },
    "spatial_localizer": {
        "module": "spatial_truncate",
        "params": [{
            "module": "spatial_localize",
            "params": [$(a relationship word), {"module": "locate_obj", "params": [$(description of an object)]}]
        }]
    }
}
```

Content within $() is what you need to consider carefully. They are not necessarily directly from the origin question: sometimes you need to think deep into the question and try to think of a reasonable sentence or word to fill in $() part.

If you think there are no temporal event that needs to be specifically located, just simplify the format as:

```
{
    "temporal_localizer": {
        "module": "temporal_truncate",
        "params": []
    },
    "spatial_localizer": {
        "module": "spatial_truncate",
        "params": [{
            "module": "spatial_localize",
            "params": [$(a relationship word), {"module": "locate_obj", "params": [$(description of an object)]}]
        }]
    }
}
```

*Figure 5.* Prompt for Question Parser (Part I).

```
Also, if you think there are no spatial area that needs to be specifically located, just simplify the format
as:
{
    "temporal_localizer": {
        "module": "temporal_truncate",
        "params": [{
            "module": "temporal_localize",
            "params": [$(a preposition), {"module": "detect_act", "params": [$(description of an event)]}]
        }]
    },
    "spatial_localizer": {
        "module": "spatial_truncate",
        "params": []
    }
}
If you think there are more than one area to be spatially located, you can add params to the
"spatial_truncate" module:
{
    "temporal_localizer": {
        "module": "temporal_truncate",
        "params": [{
            "module": "temporal_localize",
            "params": [$(a preposition), {"module": "detect_act", "params": [$(description of an event)]}]
        }]
    },
    "spatial_localizer": {
        "module": "spatial_truncate",
        "params": [{
            "module": "spatial_localize",
            "params": [$(a relationship word), {"module": "locate_obj", "params": [$(description of an
object A)]}]
        },
        {
            "module": "spatial_localize",
            "params": [$(a relationship word), {"module": "locate_obj", "params": [$(description of an
object B)]}]
        }]
    }
}
Except what I have mentioned above, you are not allowed to change the format of the policy in other
strange ways.
.
Here is some examples:
1.question: What is the boy holding after his mother entering the room?
policy:
{
    "temporal_localizer": {
        "module": "temporal_truncate",
        "params":[{
            "module": "temporal_localize",
            "params": ["after", {"module": "detect_act", "params": ["a woman enters the room"]}]
        }]
    },
    "spatial_localizer": {
        "module": "spatial_truncate",
        "params": [{
            "module": "spatial_localize",
            "params": ["holding", {"module": "locate_obj", "params": ["a boy"]}]
        }]
    }
}
```

*Figure 6.* Prompt for Question Parser (Part II).

```
2.question: Why is the girl crying?
policy:
{
    "temporal_localizer": {
        "module": "temporal_truncate",
        "params":[{
            "module": "temporal_localize",
            "params": ["when", {"module": "detect_act", "params": ["a girl is crying"]}]
        }]
    },
    "spatial_localizer": {
        "module": "spatial_truncate",
        "params": [{
            "module": "spatial_localize",
            "params": ["surrounding", {"module": "locate_obj", "params": ["a girl"]}]
        }]
    }
}
3.question: What is the animal on the left of the farmer?
policy:
{
    "temporal_localizer": {
        "module": "temporal_truncate",
        "params":[]
    },
    "spatial_localizer": {
        "module": "spatial_truncate",
        "params": [{
            "module": "spatial_localize",
            "params": ["left of", {"module": "locate_obj", "params": ["a farmer"]}]
        }]
    }
}
4.question: Which object did the person put down before they took the sandwich?
policy:
{
    "temporal_localizer": {
        "module": "temporal_truncate",
        "params": [{
            "module": "temporal_localize",
            "params": ["before", {"module": "detect_act", "params": ["the person took the sandwich"]}]
        }]
    },
    "spatial_localizer": {
        "module": "spatial_truncate",
        "params": [{
            "module": "spatial_localize",
            "params": ["putting down", {"module": "locate_obj", "params": ["the person"]}]
        }]
    }
}
```
Expand your thinking, and don't be confined to the words and phrases in the original question. Remember that your goal in decomposing the question is to temporally and spatially locate it within the video. Therefore, feel free to think creatively about how breaking down the question might help find the parts of the video that are truly relevant to the question. Based on the original question, you can make some reasonable extensions and provide appropriate policies.
Note that for the "Why" type question, it is always not enough to just locate the event and the object mentioned in the origin question. You can try to think about what events and objects might be related to the question and try to locate them.
Now I will give you a question, and you will give me the corresponding policy.

*Figure 7.* Prompt for Question Parser (Part III).

`DetectAct`. Define the UniVTG model as $M_T$, text encoder as $E_t$, video encoder as $E_v$. $V = (v_1, v_2, ..., v_T$ is an input video with $T$ frames sampled at 1fps, and $Q_a$ is a query desribing an action. We have:

$$TS = M_T(E_v(V), E_t(Q_a)), \tag{7}$$

where $TS = [t_s, t_e]$ represents a time period.

`TemporalLocalize`. $p$ is a preposition representing certain temporal relationship. $TS^{(in)} = [t_s^{(in)}, t_e^{(in)}]$ is an input time span. We have:

$$TS^{(out)} = \begin{cases} TS^{(in)}, & p \in \{\text{when, while, as}\} \\ [t_e^{(in)}, \min(2t_e^{(in)} - t_s^{(in)}, T)], & p \in \{\text{after}\} \\ [\max(0, 2t_s^{(in)} - t_e^{(in)}), t_s^{(in)}], & p \in \{\text{before}\} \end{cases} \tag{8}$$

where $TS^{(out)} = [t_s^{(out)}, t_e^{(out)}]$ is an output time span. $T$ is the video duration.

`TemporalBetween`. Given two input time spans $TS_1 = [t_{1s}, t_{1e}]$ and $TS_2 = [t_{2s}, t_{2e}]$, we have:

$$TS^{(out)} = [\min(t_{1s}, t_{2s}), \max(t_{1e}, t_{2e})], \tag{9}$$

where $TS^{(out)} = [t_s^{(out)}, t_e^{(out)}]$ is an output time span.

`TemporalTruncate`. Given an input video $V = (v_1, v_2, ..., v_T)$ and a time span $TS = [t_s, t_e]$, define $s = \lfloor t_s \rfloor, e = \lceil t_e \rceil$. We get $I = (I_1, I_2, ...I_n)$, where:

$$I_i = v_{(s + \frac{(e-s)(i-1)}{n-1})}, \tag{10}$$

and $n$ denotes the number of sampled frames.

`LocateObj`. Given the YOLO-World model $M_S$, an input image $I$, a query of an object $Q_o$, and an image encoder $E_I$ together with a text encoder $E_T$, we have:

$$BBOX = M_S(E_I(I), E_T(Q_o)), \tag{11}$$

where $BBOX = (x_1, y_1, x_2, y_2)$ is an output bounding box.

`SpatialLocalize`. Given an input image $I$, an input bounding box $BBOX^{(in)} = (x_1, y_1, x_2, y_2)$ and a word or phrase representing certain spatial relationship $r$, we have:

$$BBOX^{(out)} = \begin{cases} [\max(0, 2x_1 - x_2), y_1, x_1, y_2], & p \in \{\text{left}\} \\ [x_2, y_1, \min(w, 2x_2 - x_1), y_2], & p \in \{\text{right}\} \\ [x_1, y_2, x_2, \min(h, 2y_2 - y_1)], & p \in S_{\text{down}} \\ [x_1, \max(0, 2y_1 - y_2), x_2, y_1], & p \in S_{\text{up}} \\ [\max(0, 2x_1 - x_2), \max(0, 2y_1 - y_2), \\ \min(w, 2x_2 - x_1), \min(h, 2y_2 - y_1)], & p \in S_{\text{surround}} \end{cases} \tag{12}$$

where $S_{\text{down}} = \{\text{bottom, down, below, under, beneath, sit on, stand on, lying on}\}$, $S_{\text{up}} = \{\text{top, above, up, carry, lift, on}\}$, $S_{\text{surround}} = \{\text{next to, beside, near, surround}\}$. Particularly, if $p \in \{\text{hold, touch, contact, take}\}$, we have:

$$BBOX^{(out)} = \texttt{SpatialLocalize}(I, BBOX^{(\text{hand})}, \text{"surround"}), \tag{13}$$

where $BBOX^{(\text{hand})} = \texttt{LocateObj}(I, \text{"hand"})$.

`SpatialTruncate`. Given an input image I and a list of bounding boxes $BBOX_1, BBOX_2, ...$ where $BBOX_i = (x_{i1}, y_{i1}, x_{i2}, y_{i2})$, we have:

$$I^{(out)} = RESIZE_I(I[\min_i x_{i1}, \min_i y_{i1}, \max_i x_{i2}, \max_i y_{i2}]), \tag{14}$$

where $RESIZE_I(I')$ is the operation that resizes an image $I'$ into the shape of $I$.

## D. Detailed Alternate Self-reflection Training Strategy

The detail of our Alternate Self-reflection Training Strategy is demonstrated in Algorithm 1.

---

**Algorithm 1** Alternate Self-reflection Training Strategy

---

1: **Input:** Question parser $Q$, MoST-Grounding module $\mathcal{M}$, **temporal localizer** $\mathcal{F}_t$, **spatial localizer** $\mathcal{F}_s$, modules in **temporal localizer** $\{m_i^t\}_{i=1}^4$, modules in **spatial localizer** $\{m_i^s\}_{i=1}^3$, Multimodal LLM $\mathcal{F}$, instruction prompt $Pt$, dataset $D = \{(v_i, q_i, y_i)\}_{i=1}^N$, total training steps $S$, gradient accumulate step $s$, alternate training period $P$
2: **Initialize:** $Q, \mathcal{M}, CACHE$
3: **Freeze:** $\mathcal{M}, \{m_i^t\}, \{m_i^s\}, Q$, **Activate:** $\mathcal{F}$
4: **for** $t = 1, \ldots, S$ **do**
5:     **for** $j = 1, \ldots, s$ **do**
6:         $i \leftarrow ((t-1)s + j - 1)\%N + 1$, Prepare data $(v_i, q_i, y_i)$, Derive global representation $g_v$.
7:         Generate policy $p = Q(q_i)$
8:         Set $\mathcal{F}_{t|p}(\cdot) = \mathcal{I}(\{m_i^t\}, p)(\cdot), \mathcal{F}_{s|p}(\cdot) = \mathcal{I}(\{m_i^s\}, p)(\cdot)$, derive $c_t, c_s$ from $p$
9:         $\mathcal{M}$ **execution:** $v_s, b_{v_s} = \mathcal{M}(v_i, c_t, c_s) = \mathcal{F}_s(\mathcal{F}_t(v, c_t), c_s)$
10:         $\mathcal{F}$ **forward propagation:** $\hat{y}(q_i, v_i) = \mathcal{F}(Pt(q_i, v_s, b_{v_s}, g_v))$
11:         **Optimize** $\mathcal{F}$ **with loss:** $\mathcal{L}_{CE}$ in Equation (5)
12:         Add $(v_i, q_i, y_i)$ to $CACHE$
13:     **end for**
14:     **if** $t\%P = 0$ **then**
15:         Freeze $\mathcal{F}$, activate $Q$, initialize $\pi_\theta = Q, \pi_{\text{ref}} = Q$
16:         **for** $i = 1, \ldots, sP$ **do**
17:             Prepare data $(v_i, q_i, y_i)$
18:             Generate policies $p_1, p_2 = Q(q_i)$
19:             Forward propagation to get $\mathcal{L}_{CE1}, \mathcal{L}_{CE2}$ for $p_1, p_2$ respectively
20:             **if** $\mathcal{L}_{CE1} < \mathcal{L}_{CE2}$ **then**
21:                 $p_w \leftarrow p_1, p_l \leftarrow p_2$
22:             **else**
23:                 $p_w \leftarrow p_2, p_l \leftarrow p_1$
24:             **end if**
25:             **Optimize** $\pi_\theta$ **with loss:** $\mathcal{L}_{DPO}$ in Equation (6)
26:         **end for**
27:         $Q \leftarrow \pi_\theta$, **clear** $CACHE$, freeze $Q$, activate $\mathcal{F}$
28:     **end if**
29: **end for**

---

## E. More Visualization Examples of MSR-ViR Reasoning

Here we present some more visualization results on inference examples of our MSR-ViR framework in Figure 8 and Figure 9.

## F. Multimodal LLM Input Format

The specific input of our Multimodal LLM is illustrated in Figure 10 together with our instruction prompt. Global representation tokens are encoded and aligned global video representation $g_v$. Similarly, temporal-grounded video tokens and spatial-grounded video tokens are encoded and aligned video segments $v_s$ and bounding boxes $b_{v_s}$ respectively. Special tokens glob, tp and sp are designed to help the Multimodal LLM understand different types of tokens.

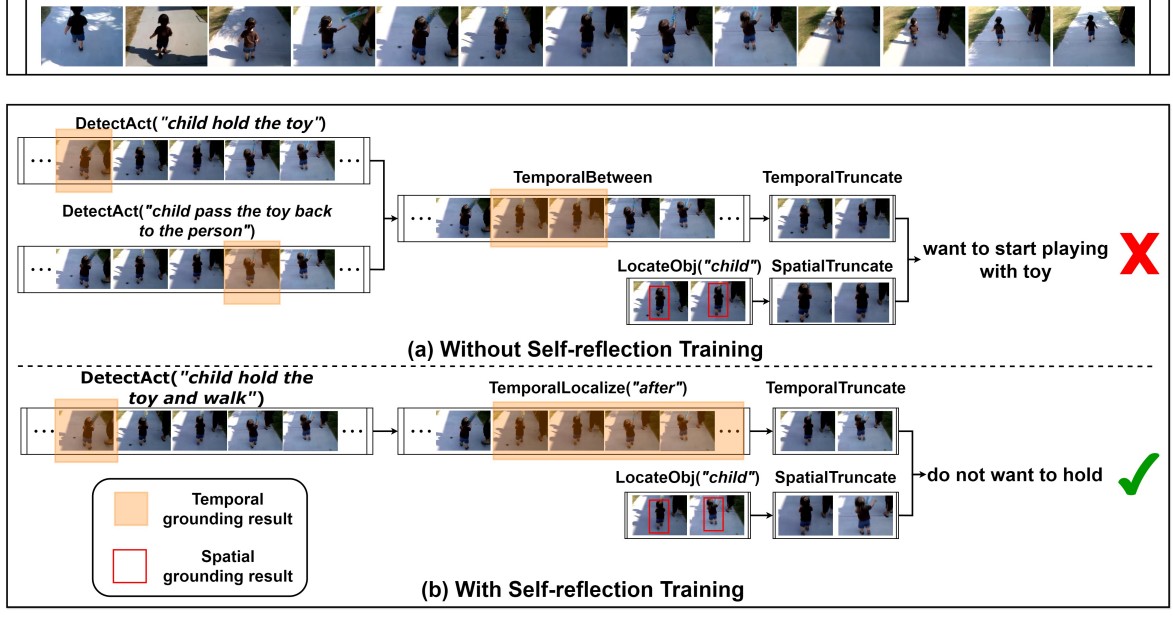

*Figure 8.* MSR-ViR inference example 1.

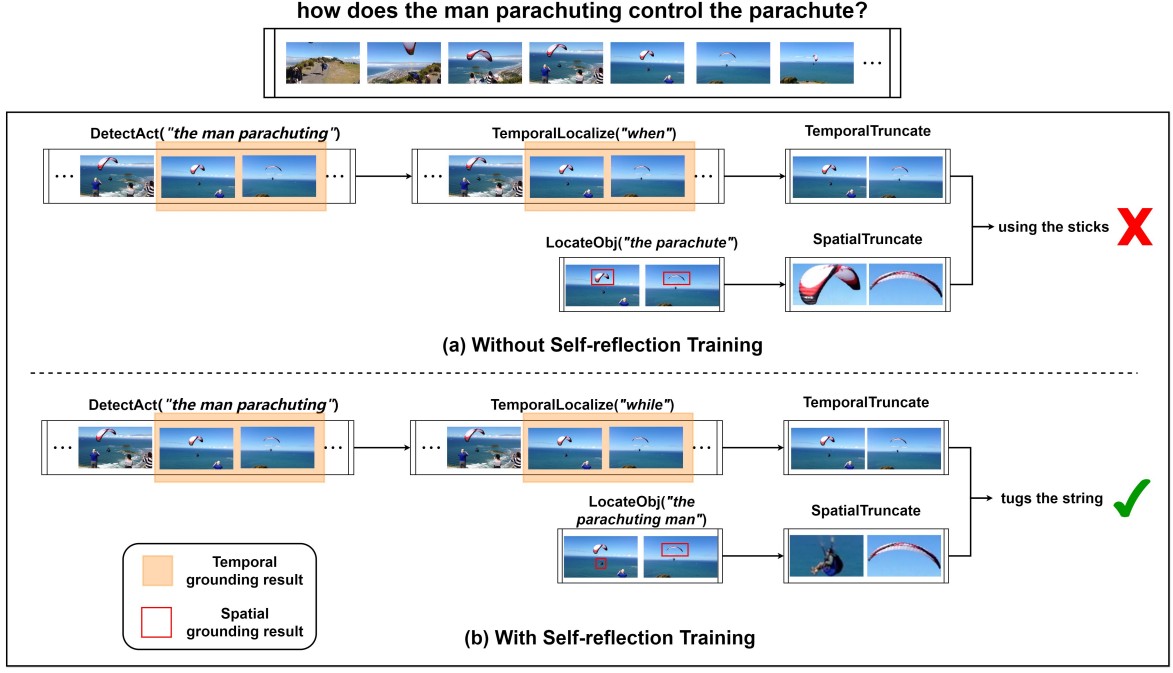

*Figure 9.* MSR-ViR inference example 2.

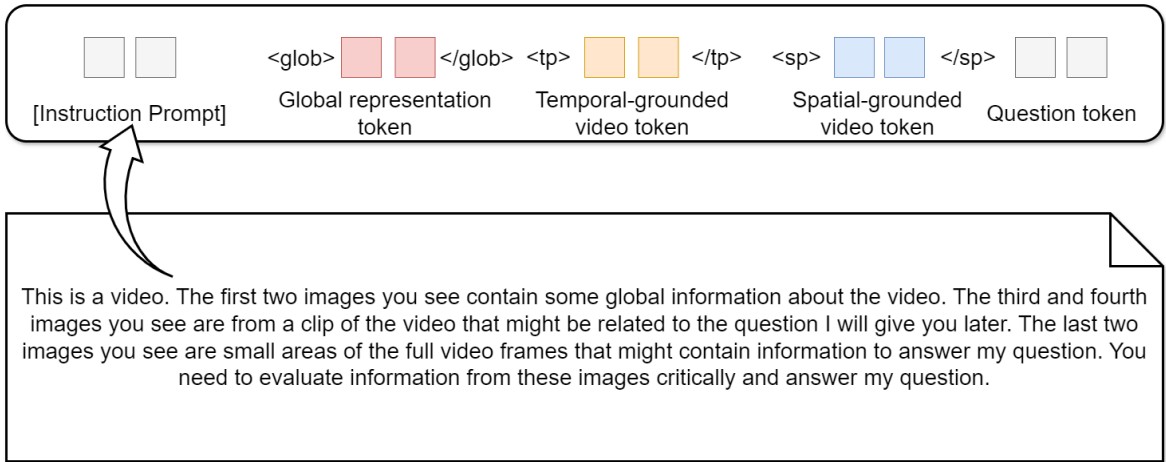

*Figure 10.* Specific input of our Multimodal LLM and the instruction prompt.

## G. More Experiments

### G.1. Complete Results on NExT-GQA

Table 7 presents the complete results on NExT-GQA dataset. Our framework MSR-ViR achieves best results on IoP, IoU and Acc@GQA.

*Table 7.* Complete results of the experiments on **NExT-GQA**. **Bold** number denotes the best result excluding gray methods.

| Method | mIoP | IoP@0.3 | IoP@0.5 | mIoU | IoU@0.3 | IoU@0.5 | Acc@GQA |
|---|---|---|---|---|---|---|---|
| VGT | 25.3 | 26.4 | 25.3 | 3.0 | 3.6 | 1.7 | 14.4 |
| VIOLETv2 | 23.6 | 25.1 | 23.3 | 3.1 | 4.3 | 1.3 | 12.8 |
| Temp[CLIP] | 25.7 | 31.4 | **25.5** | 12.1 | 17.5 | 8.9 | 16.0 |
| FrozenBiLM | 24.2 | 28.5 | 23.7 | 9.6 | 13.5 | 6.1 | 17.5 |
| TGB | - | - | - | 19.9 | 23.3 | 11.2 | - |
| LangRepo | 20.3 | - | 20.0 | 8.7 | - | 6.0 | 11.2 |
| SeViLa | 29.5 | 34.7 | 22.9 | 21.7 | 29.2 | 13.8 | 16.6 |
| LLoVi(Mistral-7B) | 20.7 | - | 20.5 | 8.7 | - | 6.0 | 11.2 |
| LLoVi(GPT-4) | 37.3 | - | 36.9 | 20.0 | - | 15.3 | 24.3 |
| MoReVQA(Palm-2) | 37.8 | - | 37.6 | 19.7 | - | 15.4 | 39.6 |
| MSR-ViR$_Q$(ours) | **30.0** | **39.8** | 25.0 | 22.8 | 33.0 | **16.4** | 18.5 |
| MSR-ViR$_L$(ours) | 29.6 | 39.0 | 24.1 | **23.4** | **33.6** | **16.4** | **18.6** |

### G.2. Experiments on Computational Efficiency of MSR-ViR

For further comparison between our MSR-ViR framework and baselines, including end-to-end Multimodal LLMs and other grounding-based methods, we conduct experiments on NExT-QA to test their inference speed, and the results, together with parameter size and accuracy on NExT-QA, are demonstrated in Table 8. The inference speed is tested on one NVIDIA A100 GPU. Our framework's additional parameters mainly stem from the Question Parser, and MoST-Grounding contributes less than 0.1B. For comprehensive comparison, we test total parameters, inference speed, and accuracy using Question Parsers of different sizes (Qwen2-7B and Qwen2-1.5B). With Qwen2-7B Question Parser, the inference speed of MSR-ViR is about twice that of the direct baseline, consistent with our complexity estimates. With Qwen2-1.5B Question Parser, although accuracy slightly drops, it still outperforms the direct baseline with fewer additional parameters and less computational overhead.

*Table 8.* Model size and inference speed on NExT-QA dataset. * denotes the method is designed for high-efficiency inference.

| Method | Parameter Size | Inference Speed | Acc on NExT-QA |
|---|---|---|---|
| BLIP-2 | 4.1B | 1.21 s / sample | 69.6 |
| TGB* | 4.3B | 1.62 s / sample | 72.1 |
| InstructBLIP | 7.9B | 1.75 s / sample | 72.5 |
| SeViLa | 8.3B | 2.79 s / sample | 73.8 |
| Qwen-VL | 9.6B | 1.32 s / sample | 71.9 |
| MSR-ViR$_Q$(1.5B parser) | 11.2B | 2.35 s / sample | 73.1 |
| MSR-ViR$_Q$(7B parser) | 16.7B | 3.10 s / sample | 73.6 |
| LLaVA-NeXT | 7.1B | 2.19 s / sample | 73.1 |
| MSR-ViR$_L$(1.5B parser) | 8.7B | 4.29 s / sample | 74.2 |
| MSR-ViR$_L$(7B parser) | 14.2B | 4.96 s / sample | 74.9 |

### G.3. Ablation Study on Grounding Model

In MoST-Grounding module, we utilize a small grounding model UniVTG as our temporal grounding module. To demonstrate the effectiveness of UniVTG, we further conduct ablation study on the choice of temporal grounding model. We utilize $R^2$-Tuning (Liu et al., 2024) and Moment-DETR (Lei et al., 2021) to replace UniVTG and test on NExT-GQA dataset, and the results are shown in Table 9. MSR-ViR$_Q$ with UniVTG achieves the best results on NExT-GQA.

*Table 9.* Ablation study for temporal grounding models on NExT-GQA. This is the test result of MSR-ViR$_Q$ with different temporal grounding models UniVTG, $R^2$-Tuning and Moment-DETR.

| Grounding Model | Acc@QA | Acc@GQA | mIoP | IoP@0.5 | mIoU | IoU@0.5 |
|---|---|---|---|---|---|---|
| UniVTG | **69.9** | **18.5** | **30.0** | **25.0** | **22.8** | **16.4** |
| $R^2$-Tuning | 67.3 | 16.6 | 28.7 | 23.2 | 22.7 | 15.9 |
| Moment-DETR | 67.4 | 17.2 | 28.6 | 24.1 | 21.4 | 14.7 |

### G.4. Ablation Study on Frame Sampling Strategy

In Section 4.1, we provide detailed implementations of MSR-ViR and its direct baseline, including the leveraged frame sampling strategy. Here we present further ablation study on frame sampling strategy for Qwen-VL, LLaVA-NeXT, MSR-ViR$_Q$ and MSR-ViR$_L$ on NExT-QA dataset to show that our implementations are optimized for each model. The results are shown in Figure 11. We utilize the sampling strategy with the highest accuracy on NExT-QA. It could also be seen that with the same number of sampled frames, MSR-ViR significantly outperforms its direct baseline, further demonstrating the effectiveness of our proposed framework.

## H. Upper Bound Computational Complexity of MSR-ViR

To enhance interpretability of Multimodal LLMs, MSR-ViR incorporates several additional models (as illustrated in Section 3.1 and Appendix C), thereby elevating the computational complexity of the reasoning framework. For comprehensive evaluation, we furnish the upper bound computational complexity of our reasoning framework MSR-ViR$_Q$. With this upper bound, we demonstrate that the computational complexity of our reasoning framework lies within a reasonable and controllable range. As transformers are widely utilized in our framework, we first furnish the proof of the following lemma:

**Lemma H.1.** *For a given transformer with $N_L$ layers, hidden size $d$, intermediate size $d_{ff}$ (in feed forward networks) and a given input with $l$ tokens, the computational complexity of inference is $O\left(N_L\left(4d^2l + 2dl^2 + 2dd_{ff}l\right)\right)$.*

*Proof.* The computational time bottlenecks of a transformer lie in the multi-head self-attention and feed-forward network, so we calculate the computational complexity of these two components respectively.

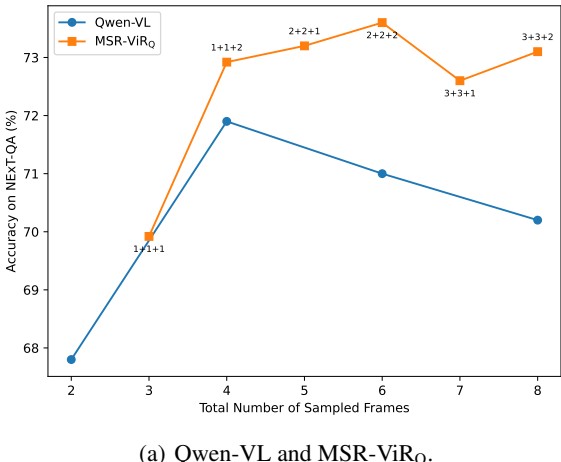

(a) Qwen-VL and MSR-ViR$_Q$.

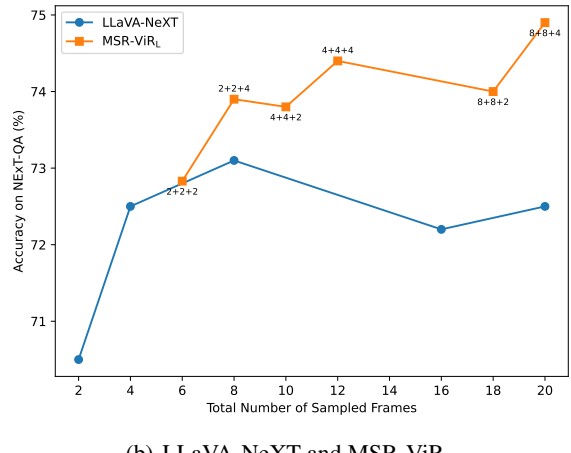

(b) LLaVA-NeXT and MSR-ViR$_L$.

*Figure 11.* Ablation study on frame sampling strategy on NExT-QA. For MSR-ViR, the number in the figures represents (temporal-grounded frames + spatial-grounded frames + global representation frames).

In multi-head self-attention, the calculation of the $Q, K, V$ matrices is required first, which involves matrix multiplication between a matrix of shape $l \times d$ and a matrix of shape $d \times d$. So the computational complexity of this process is $O\left(3d^2l\right)$.

Next, assume the number of heads in multi-head self-attention is $h$. So the dimension of each attention head will be $d_k = d/h$. For each attention head:

$$\text{Attention}(Q, K, V) = \text{softmax}\left(\frac{QK^{\mathrm{T}}}{\sqrt{d_k}}\right) V.$$

So the computational complexity of one attention head is $l^2 d_k + l^2 d_k = 2l^2 d_k$. For $h$ heads, the overall computational complexity will be $2l^2 d_k \times h = 2l^2 d$. The final linear layer involves the matrix multiplication between a matrix of shape $l \times d$ and a matrix of shape $d \times d$, thus the computational complexity is $ld^2$. Overall, the computational complexity of multi-head self-attention is $3d^2l + 2dl^2 + d^2l = 4d^2l + 2dl^2$.

The feed-forward network involves two linear layers, the hidden size of which is $d_{ff}$, so the computational complexity of feed-forward network is $2dd_{ff}l$. With $N_L$ layers, the computational complexity will be multiplied by $N_L$. Up to this point, we have proved Lemma H.1: the computational complexity of a transformer is $O\left(N_L\left(4d^2l + 2dl^2 + 2dd_{ff}l\right)\right)$.                $\square$

Next, we will calculate the computational complexity of each modules in MSR-ViR$_Q$. For the Question Parser Qwen2-7B:

**Proposition H.2.** *Given an input with $l$ tokens, the computational complexity of Qwen2-7B is $O\left(P_1l^2\right)$, where $P_1$ is a constant relevant to parameters of Qwen2-7B.*

*Proof.* Trivially, the computational bottlenecks of Qwen2-7B lie in transformers. According to Lemma H.1, the computational complexity of Qwen2-7B is $O\left(N_L\left(4d^2l + 2dl^2 + 2dd_{ff}l\right)\right)$. Considering $d$, $N_L$ and $d_{ff}$ as constants and eliminating low-order terms, the complexity expression of Qwen2-7B could be simplified as $O\left(P_1l^2\right)$.                $\square$

**Proposition H.3.** *Given a video with length $T$ and resolution $H \times W$, and a query with $l_q$ tokens, the computational complexity of UniVTG is $O\left(P_2TH^2W^2 + P_3T^2 + P_4l_q^2 + P_5Tl_q\right)$, where $P_2$, $P_3$, $P_4$ and $P_5$ are constants relevant to parameters of UniVTG.*

*Proof.* UniVTG incorporates a CLIP model, which involves a visual encoder and a text encoder, to encode the input videos and queries. Then a multimodal encoder is utilized for cross-modal interaction. All sub-modules of UniVTG are based on transformers. We will calculate the computational complexity of the three encoders respectively. Assume the parameters of transformers in the visual encoder are $d_V, L_V, d_{ff_V}$, the parameters of transformers in the text encoder are $d_T, L_T, d_{ff_T}$ and the parameters of transformers in the multimodal fusion encoder are $d_F, L_F, d_{ff_F}$. (The parameters of transformers are as illustrated in Lemma H.1.)

For the visual encoder, the image is first input into a convolution layer Conv2d(3, $d_V$, kernel_size=$(s, s)$, stride=$(s, s)$)). The complexity of this step would be $\frac{HW}{s^2} \times s \times s \times 3 \times d_V = 3d_V HW$. Then the encoded image of shape $\frac{HW}{s^2} \times d_V$ is fed into a vision transformer. Let $n_V = \frac{HW}{s^2}$ be the number of visual tokens. According to Lemma H.1, the computational complexity of the vision transformer would be $L_V(4n_V d_V^2 + 2n_V^2 d_V + 2n_V d_V d_{ff_V})$. For a video with length $T$, the total complexity of the visual encoder would be $O\left(T\left(3d_V HW + L_V(4n_V d_V^2 + 2n_V^2 d_V + 2n_V d_V d_{ff_V})\right)\right)$. For the text encoder, given the length of query $l_q$, the computational complexity would be $O\left(L_T\left(4l_q d_T^2 + 2l_q^2 d_T + 2l_q d_T d_{ff_T}\right)\right)$ according to Lemma H.1.

For the multimodal encoder, given the length of input (including visual tokens and text tokens) $n = T + l_q$, the computational complexity would be $O\left(L_F\left(4n d_F^2 + 2n^2 d_F + 2n d_F d_{ff_F}\right)\right)$ according to Lemma H.1.

Overall, the computational complexity of UniVTG would be: $O(T(3d_V HW + L_V(4n_V d_V^2 + 2n_V^2 d_V + 2n_V d_V d_{ff_V})) + L_T(4l_q d_T^2 + 2l_q^2 d_T + 2l_q d_T d_{ff_T}) + L_F(4(T+l_q)d_F^2 + 2(T+l_q)^2 d_F + 2(T+l_q)d_F d_{ff_F}))$ where $n_V = \frac{HW}{s^2}$. Considering all the parameters as constants and eliminating low-order terms, the complexity expression of UniVTG could be simplified as $O\left(P_2 T H^2 W^2 + P_3 T^2 + P_4 l_q^2 + P_5 T l_q\right)$.

$\square$

**Proposition H.4.** *Given an image with shape $H \times W$, and a query with length $l_q$, the computational complexity of YOLO-World is $O\left(P_6 HW + P_7 l_q^2\right)$, where $P_6$ and $P_7$ are constants relevant to parameters of YOLO-World.*

*Proof.* YOLO-World consists of multiple convolution structures. Since the calculation process of these structures is repetitive and cumbersome, the detailed calculation process of the computational complexity of YOLO-World is omitted here. Evidently, the complexity of the convolution calculation is proportional to $HW$, and the complexity of the text encoding calculation is proportional to $l_q^2$ (according to Lemma H.1). So the computational complexity of YOLO-World is approximately $O\left(P_6 HW + P_7 l_q^2\right)$.

$\square$

**Proposition H.5.** *Given the number of input frames $N$, the shape of input frames $H \times W$ and the length of text input $l$, the computational complexity of Qwen-VL-7B is $O\left(P_8 N H^2 W^2 + P_9 N^2 + P_{10} l^2 + P_{11} N l\right)$, where $P_8$, $P_9$, $P_{10}$ and $P_{11}$ are constants relevant to parameters of Qwen-VL-7B.*

*Proof.* Qwen-VL utilizes a vision transformer to encode the input images, after which a cross attention layer with learnable query embeddings is used to project the visual tokens into the space of text tokens. Finally QwenLM gives the output according to the input visual tokens and text tokens. Assume the parameters of the vision transformer are $d_{VT}, L_{VT}, d_{ff_{VT}}$, the parameters of QwenLM are $d_Q, L_Q, d_{ff_Q}$. $n_q$ is the number of queries in the cross attention layer.

As we have mentioned in the proof of Proposition H.3, the computational complexity of the vision transformer would be $O\left(3d_{VT} HW + L_{VT}(4n_{VT} d_{VT}^2 + 2n_{VT}^2 d_{VT} + 2n_{VT} d_{VT} d_{ff_{VT}})\right)$, where $n_{VT} = \frac{HW}{s^2}$ is the number of visual tokens and $s$ is the kernel size and stride in the convolution layer.

The calculation of the computational complexity of the cross attention layer is similar to the proof of Lemma H.1, which is omitted here. The computational complexity of the cross attention layer is $O\left(n_{VT} d_{VT} d_Q + 2n_q n_{VT} d_Q + 2n_q d_Q^2 + 2n_{VT} d_Q^2\right)$. $n_{VT} d_{VT} d_Q$ is the complexity of the linear layer before cross-attention, projecting the visual tokens in the space of dimension $d_{VT}$ to the space of dimension $d_Q$.

Finally for QwenLM, the total length of the input would be $n_Q = N n_q + l$. According to Lemma H.1, the computational complexity of QwenLM would be $L_Q(4n_Q d_Q^2 + 2n_Q^2 d_Q + 2n_Q d_Q d_{ff_Q})$.

Overall, the computation complexity of Qwen-VL would be $O(N(3d_{VT} HW + L_{VT}(4n_{VT} d_{VT}^2 + 2n_{VT}^2 d_{VT} + 2n_{VT} d_{VT} d_{ff_{VT}}) + n_{VT} d_{VT} d_Q + 2n_q n_{VT} d_Q + 2n_q d_Q^2 + 2n_{VT} d_Q^2) + L_Q(4(N n_q + l)d_Q^2 + 2(N n_q + l)^2 d_Q + 2(N n_q + l)d_Q d_{ff_Q}))$, where $n_{VT} = \frac{HW}{s^2}$. Considering all the parameters of the model as constants and eliminating low-order terms, the complexity of Qwen-VL would be $O\left(P_8 N H^2 W^2 + P_9 N^2 + P_{10} l^2 + P_{11} N l\right)$.

$\square$

**Proposition H.6.** *Given a video with length $T$ and resolution $H \times W$, length of our prompts for the Question Parser $l_p$ and a question with length $L$, the computational complexity of MSR-ViR$_Q$ is $O\left(c_1 T H^2 W^2 + c_2 T^2 + C_1 H^2 W^2 + C_2 L^2 + C_3 l_p^2\right)$,*

*where $c_1$, $c_2$ are constants relevant to parameters of small models in MSR-ViR$_Q$, and $C_1$, $C_2$ and $C_3$ are constants relevant to parameters of large models in MSR-ViR$_Q$.*

*Proof.* All other modules only involve operations with a complexity of $O(1)$, such as resizing and cropping. Therefore, only the above four parts (Question Parser, UniVTG, YOLO-World, Qwen-VL) need to be considered when calculating the complexity of MSR-ViR$_Q$. When MSR-ViR$_Q$ answers a question, it invokes the Question Parser and Qwen-VL for one time respectively. In MoST-Grounding, considering the most complex policy where `TemporalBetween` module is utilized to invoke UniVTG twice, while in the **spatial_localizer**, YOLO-World needs to be invoked twice for each image for spatial localization.

Let the complexity of the Question Parser be $T_P$, the complexity of UniVTG be $T_U$, the complexity of YOLO-World be $T_Y$ and the complexity of Qwen-VL be $T_Q$. We sample $N$ frames as temporal-grounded frames, $N$ frames as spatial-grounded frames. The length of the question is $L$, and the length of the query for UniVTG and YOLO-World is $l_q$. The length of the video is $T$, and the resolution of which is $H \times W$. Then, the upper bound complexity of Qwen-VL would be:

$$
\begin{aligned}
T &= T_P + 2T_U + 2NT_Y + T_Q \\
&= P_1(L + l_p)^2 + 2P_2TH^2W^2 + 2P_3T^2 + 2P_4l_q^2 + 2P_5Tl_q + 2NP_6HW + 2NP_7l_q^2 \\
&\quad + 2P_8NH^2W^2 + 4P_9N^2 + P_{10}L^2 + 2P_{11}NL,
\end{aligned}
$$

It is worth noting that $l_q$ is usually the length of a very short sentence, thus the terms with $l_q$ could be eliminated. Also notice that the length of question $L$ is significantly smaller than the length of prompts $l_p$. After eliminating low-order terms:

$$
\begin{aligned}
T &= P_1(L + l_p)^2 + 2P_2TH^2W^2 + 2P_3T^2 + 2NP_6HW + 2P_8NH^2W^2 + 4P_9N^2 + P_{10}L^2 + 2P_{11}NL \\
&\approx P_1l_p^2 + P_{10}L^2 + 2P_2TH^2W^2 + 2P_3T^2 + 2P_8NH^2W^2 \\
&= c_1TH^2W^2 + c_2T^2 + C_1H^2W^2 + C_2L^2 + C_3l_p^2,
\end{aligned}
$$

where $c_1 = 2P_2$, $c_2 = 2P_3$ are constants relevant to parameters of the small model UniVTG, and $C_1 = 2NP_8$, $C_2 = P_{10}$ and $C_3 = P_1$ are constants relevant to parameters of the large models Qwen2-7B and Qwen-VL-7B. (We set $N = 2$ in MSR-ViR$_Q$, so actually $C_1 = 4P_8$).

$\square$

According to Proposition H.6, the upper bound computational complexity of MSR-ViR$_Q$ is $O(c_1TH^2W^2 + c_2T^2 + C_1H^2W^2 + C_2L^2 + C_3l_p^2)$. As $c_1$ and $c_2$ are constants from a small model, they are significantly smaller than $C_1$ and $C_2$. So the execution time of MoST-Grounding, which is approximately $O\left(c_1TH^2W^2 + c_2T^2\right)$, would be strictly bounded, although it is proportional to $TH^2W^2$ and $T^2$. According to Proposition H.5, if we fix $N$ as a constant, complexity for Qwen-VL would be $O(P_{12}H^2W^2 + P_{10}L^2)$. After comparison, it is obvious that the computational overhead of our framework mostly comes from $C_3l_p^2$, as $l_p$ is significantly larger than $L$.

In another word, the Question Parser is the main source of computational overhead. This is acceptable for our framework, as the modular policies generated by the Question Parser provide interpretable reasoning path. Besides, this part of computational complexity is irrelevant to the length and resolution of the video, and the length of question itself, making it strictly bounded no matter how complex the scenario is. According to the complexity calculation results, the inference time of our framework should be approximately 2 to 3 times that of the baseline. The experimental results in Table 8 verify the correctness of this. Overall, the additional computational overhead introduced by our framework is reasonable and strictly bounded.

