# OpenReview forum: "Modularized Self-Reflected Video Reasoner for Multimodal LLM with Application to Video Question Answering"
_ICML.cc/2025/Conference — ICML 2025 poster_

### Official Review · Reviewer_u3Mw · 2025-03-07

**Overall Recommendation:** 3

**Summary:**

This paper enhances the interpretability and reasoning capabilities of Multimodal Large Language Models (MLLMs) in video question answering.
The authors propose a modular system that constructs explicit reasoning paths and extracts precise spatial-temporal information.
The framework is further optimized using a Reinforcement Learning-based self-reflection mechanism.
Evaluations on STAR, NExT-QA, and other VideoQA benchmarks show that the MoST-Grounding network achieves superior performance and interpretability.

**Claims And Evidence:**

Yes. The claims are clearly proved.

**Essential References Not Discussed:**

Q6. The modular or agentic system, which decomposes complex questions into simpler sub-questions and extracts spatial-temporal information for better VideoQA performance and interpretability, has also been explored in several prior works, including STAIR [1], VideoAgent [2], ProViQ [3], AoTD [4], ENTER[5] and MotionEpic [6].

---
[1] STAIR: Spatial-Temporal Reasoning with Auditable Intermediate Results for Video Question Answering. Yueqian Wang, et al. AAAI 2024.

[2] VideoAgent: A Memory-augmented Multimodal Agent for Video Understanding. Yue Fan, et al. ECCV 2024.

[3] Video Question Answering with Procedural Programs. Rohan Choudhury, et al. ECCV 2024.

[4] Enhancing Video-LLM Reasoning via Agent-of-Thoughts Distillation. Yudi Shi, et al. CVPR 2025.

[5] ENTER: Event Based Interpretable Reasoning for VideoQA. Hammad Ayyubi, et al. Arxiv, 2024.

[6] Video-of-Thought: Step-by-Step Video Reasoning from Perception to Cognition. Hao Fei, et al. ICML 2024.

**Experimental Designs Or Analyses:**

Q4. **Frame Sampling:** LLaVA-NeXT uniformly samples 8 frames, Qwen-VL samples 4 frames, while MSR-ViR Q samples 4 frames for spatial-temporal information and 2 for global information, and MSR-ViR L samples 16 frames for spatial-temporal information and 2 for global information. Does this unequal sampling strategy introduce bias in performance comparisons? Clarifying whether the frame sampling rates are optimized for each model or standardized for fair evaluation would strengthen the validity of the results.

Q5. **Base Models:** The base models, LLaVA-NeXT and Qwen-VL, appear outdated. Replacing them with current mainstream models, such as LLaVA-Video or Qwen2-VL, would provide a more convincing and up-to-date comparison, reflecting the latest advancements in the field.

**Methods And Evaluation Criteria:**

Below are some concerns about the proposed method.

Q1. **Lack of Novelty**: While the authors claim that MSR-ViR differs from grounding-based methods and modular approaches like MoReVQA, its core methodology—decomposing questions, performing spatial-temporal grounding, and using a model (LLM or MLLM) to generate answers—has been extensively explored in prior works such as STAIR [1] and VideoAgent [2]. The substitution of the final answering model adds limited novelty. Although the self-reflection learning method introduces a new perspective, the overall improvements appear limited.

Q2. **Self-Reflection Learning for Multiple-choice QAs**: The authors use DPO in the self-reflection process, treating policies with smaller losses as positive. Is the loss referred to here the SFT loss? If so, for multiple-choice data, such as NExT-QA and STAR-sub, various polices may yield the same choice and the same loss. How do the authors distinguish between positive and negative samples in such cases? Please clarify if my understanding is incorrect.

Q3. **Handling Non-Spatial-Temporal Questions**: How does MSR-ViR address questions that do not require spatial-temporal information but still demand interpretable answers, such as, "What makes the video humorous?"

---
[1] STAIR: Spatial-Temporal Reasoning with Auditable Intermediate Results for Video Question Answering. Yueqian Wang, et al. AAAI 2024.

[2] VideoAgent: A Memory-augmented Multimodal Agent for Video Understanding. Yue Fan, et al. ECCV 2024.

**Other Comments Or Suggestions:**

None.

**Other Strengths And Weaknesses:**

**Strength:**

1. The paper is well-written, with a clear motivation and logical flow, making it easy to follow.

2. The use of Reinforcement Learning for the question parser effectively circumvents the challenge of supervising intermediate reasoning steps, aligning with contemporary approaches like Deepseek-R1.

**Weaknesses:**

See above.

**Questions For Authors:**

See above. I would consider raising my score if the authors address my concerns.

**Relation To Broader Scientific Literature:**

This paper distinguishes itself from conventional MLLMs, which typically generate answers directly without intermediate reasoning processes. By incorporating the MoST-Grounding module, the proposed approach provides the model with more precise spatial-temporal information, thereby enhancing the interpretability of the QA process and improving the overall performance of the model.

**Theoretical Claims:**

What does $P_4$ mean in Proposition 3.1?

---

> ### Author Rebuttal · Authors · 2025-03-31
>
> We sincerely thank the reviewer for taking time to review our paper and providing insightful feedback and suggestions. We address the weaknesses and questions as follows:
>
> ### **Methods And Evaluation Criteria**
>
> #### **Q1 Lack of Novelty**
>
> We highlight the contributions of our paper as follows:
>
> MSR-ViR enhances interpretability in Multimodal LLMs for VideoQA by integrating a modular network for interpretable reasoning. An alternate self-reflection training strategy is proposed wherein Multimodal LLM provides feedback to perform DPO training on Question Parser that generates reasoning paths. To the best of our knowledge, this is the first work that not only integrates a modular network into a Multimodal LLM but also jointly optimize them with self-reflection training for reasoning path refinement and QA accuracy improvement. Prior modular-based works(VideoAgent, AoTD) directly use the results of the modular system without optimizing or refining them, and some of them are built on Uni-modal LLM(MoReVQA, ProViQ). As for the limited performance improvement issue, we address as follows:
>
> - Grounding-based methods are not capable of significantly improving the accuracy of Multimodal LLMs in answering questions, as they basically only change the input without changing the model's internal structure. More relevant visual information helps improve QA accuracy, but the improvement is limited. This aligns with existing grounding-based approaches, where improvements remain modest (e.g., SeViLa +1.2%, GCG +2.1%). The improvement of our method is comparable to that of the existing works.
> - It can be observed that the performance improvement on EgoSchema is more significant compared to that on NExT-QA. We claim that MSR-ViR shows greater performance gains on longer video datasets, where grounding and reasoning are more essential. For further validation, we train Qwen2-VL and LLaVA-Video with baseline method and MSR-ViR on a subset of LLaVA-Video-178K and test them on NExT-QA and a longer VideoQA dataset VideoMME. The results demonstrate the superiority of MSR-ViR, especially on longer videos.
>
> |NExT-QA|Tem.| Cau.| Des.| Avg.|
> |--------------|----------|----------|----------|----------|
> |Qwen2-VL|74.6|78.2|83.1|77.8|
> |MSR-ViR$_{Q2}$|75.9(+1.3)|80.2(+2.0)|84.5(+1.4)|79.6(+1.8)|
> |LLaVA-Video|79.0|82.6|86.3|82.1|
> |MSR-ViR$_{LV}$|81.8(+2.8)|84.8(+2.2)|87.3(+1.0)|84.3(+2.2)|
>
> |VideoMME|Short|Medium|Long |Avg.|
> |--------------|----------|----------|----------|----------|
> |Qwen2-VL|65.2|52.2|48.3|55.3|
> |MSR-ViR$_{Q2}$|66.8(+1.6)|55.4(+3.2)|51.3(+3.0)|57.9(+2.6)|
> |Llava-Video|69.7|56.6|49.3|58.5|
> |MSR-ViR$_{LV}$|72.3(+2.6)|60.7(+4.1)|52.6(+3.3)|61.9(+3.4)|
>
> #### **Q2 Self-Reflection Learning for Multiple-choice QAs**
>
> The loss here refers to SFT loss. For multi-choice QAs, this loss simplifies to the correctness of the selected option. For policy pairs that yield the same option, we do not use them for DPO training. About 50% of policy pairs could be used in the first epoch, adequate for DPO training of Question Parser.
>
> #### **Q3 Handling Non-Spatial-Temporal Questions**
>
> The interpretability in our work is considered from the perspective of spatial-temporal grounding. Non-spatial-temporal questions constitute a special scenario for our framework. In these cases, MSR-ViR asserts that answering the questions necessitates the entirety of video information. In other words, the grounding result would be the full video. Future work could expand MSR-ViR with non-spatial-temporal modules for further interpretability.
>
> ### **Theoretical Claims**
>
> We are sorry about the typo. Please refer to "Theoretical Claims" part in our rebuttal to reviewer NjDQ for more details.
>
> ### **Experimental Designs Or Analyses**
>
> #### **Q4 Frame Sampling**
>
> The current frame sampling rates are optimized. Due to character limitation, we present several ablation on frame sampling for NExT-QA in this link: https://anonymous.4open.science/api/repo/frame-ablation-C4EB/file/frame-ablation.png?v=54b8a357. For MSR-ViR, the number represents (temporal + spatial + global).
>
> #### **Q5 Base Models**
>
> We conduct experiments for Qwen2-VL and LLaVA-Video on NExT-QA and VideoMME. Please refer to Q1.
>
> ### **Essential References Not Discussed**
>
> STAIR generates policies through small models trained with supervision on a single dataset, thus the types of questions it could solve are severely limited. ProViQ and ENTER utilize LLM to generate programs. However, they are based on Uni-modal LLM, and may fail to generate executable programs due to lack of training. VideoAgent and MotionEpic utilize Chain-of-Thought which involves multi-round conversation with LLM, while AoTD distills knowledge from CoT into Video LLM to improve instruction tuning. The idea of jointly optimizing Multimodal LLM and modular system via self-reflection training in our work represents a novel approach not explored in these works. We will add detailed discussion to the final version of our paper.

---

> > ### Comment · Reviewer_u3Mw · 2025-04-07
> >
> > Thanks for your reply!
> > This really solves my concerns.
> > The experiments for Qwen2-VL and LLaVA-Video show the effectiveness of the method on SOTA MLLMs.
> > So I will raise my score to 3.

---

> > > ### Author Response · Authors · 2025-04-07
> > >
> > > We sincerely thank the reviewer again for the insightful review and reply, which definitely further improve the quality of our paper. We will add the experiments and discussions to the final version of our paper.

---

### Official Review · Reviewer_uxoU · 2025-03-14

**Overall Recommendation:** 4

**Summary:**

The paper addresses the interpretability problem in VideoQA by introducing the Modularized Self-Reflected Video Reasoner (MSR-ViR) framework, which decomposes complex questions into smaller parts through its Modularized SpatialTemporal Grounding (MoST-Grounding) module and employs a reinforcement learning-based Alternate Self-reflection Training strategy to train a Multimodal LLM. By following a tree-structured execution policy from a Question Parser, MoST-Grounding progressively isolates relevant visual information from the video, improving both spatial-temporal grounding and reasoning. This yields transparent reasoning paths, as well as visual evidence for predicted answers. A theoretical analysis demonstrates bounded computational overhead, and experiments show that MSR-ViR not only outperforms baselines but also accurately localizes temporal segments, enhancing interpretability in VideoQA datasets such as EgoSchema, NExT-QA, STAR, and NExT-GQA.

**Claims And Evidence:**

Yes, the claims are supported by strong performances as well as ablations on verifying whether each design choice is necessary.

**Essential References Not Discussed:**

The authors should discuss a relevant paper from CVPR 2024: [1]

[1] Di, Shangzhe, and Weidi Xie. "Grounded question-answering in long egocentric videos." Proceedings of the IEEE/CVF Conference on Computer Vision and Pattern Recognition. 2024.

**Experimental Designs Or Analyses:**

The experimental design is comprehensive.

**Methods And Evaluation Criteria:**

There are multiple components in the method: a question parser to decompose questions into sub-questions, two modularized temporal localizer and spatial localizers to extract temporal-grounded and spatial-grounded frames. Then the frames are fed into a multimodal LLM for supervised fine-tuning, and lastly, an Alternate Self-reflection training strategy to improve the question parser by using a DPO over multiple parsing and preferring the one with the lowest multimodal LLM loss.

The benchmarks and metrics are standard and complete, ranging from accuracies of different categories in NExT-QA, STAR-sub, and EgoSchema (sub and full), as well as mIoU, IoU, Io, and accuracy of NExT-GQA.

**Other Comments Or Suggestions:**

NA

**Other Strengths And Weaknesses:**

NA

**Questions For Authors:**

NA

**Relation To Broader Scientific Literature:**

This paper leverages grounding to improve video QA accuracies while providing interpretability because of modularization, which is an important contribution compared to previous models, which mostly rely on captions or do not provide any interpretability.

**Theoretical Claims:**

The reviewer briefly checked the proofs of propositions 3.1 and 3.2 and they seem correct.

---

> ### Author Rebuttal · Authors · 2025-03-31
>
> We sincerely thank the reviewer for taking time to review our paper and providing insightful feedback and suggestions. We address the questions as follows:
>
> ### **Supplementary Material**
>
> We will open-source our code after the review and provide detailed documentation to facilitate the reproduction and application of our framework.
>
> ### **Essential References Not Discussed**
>
> The CVPR paper addresses the challenge of open-ended VideoQA in long egocentric videos. The authors propose a novel approach that integrates query grounding and answer generation into a unified model to achieve grounded VideoQA. However, the proposed unified multimodal model is still black-box, lacking interpretability. Different from this paper, our work provides interpretability in VideoQA tasks for Multimodal LLMs. We will add detailed discussion to the final version of our paper.

---

> > ### Comment · Reviewer_uxoU · 2025-04-04
> >
> > The reviewer thanks the authors with the response, which addresses the reviewer's initial concerns. The reviewer keeps a positive rating of 4.

---

> > > ### Author Response · Authors · 2025-04-07
> > >
> > > We sincerely thank the reviewer for the insightful review and suggestions, and we will add the further discussions to the final version of our paper. Also, we will open-source our code after the review to facilitate the reproduction and application of our proposed method.

---

### Official Review · Reviewer_NjDQ · 2025-03-14

**Overall Recommendation:** 3

**Summary:**

This paper introduces MSR-ViR, a novel framework designed for interpretability of video question answering. Multiple modular networks are integrated wth a multimodal large language model in the proposed method.

To refine its reasoning, the framework utilizes an Alternate Self-reflection Training Strategy, optimizing both the policy generation and the multimodal LLM.

Experiments on different datasets show that the proposed method can improve video understanding and the accuracy of localizing evidence for answers.

**Claims And Evidence:**

1. MSR-ViR provides interpretable VideoQA with explicit reasoning paths: The paper proposes the grounding modules to decompose complex questions and localizes relevant video segments. However, the tree-structured policy generated by the Question parser seems a little heavy. The full prompt is provided in the supplementary, comprising three pages.

2. MSR-ViR enhances the VideoQA abilities of Multimodal LLMs: This claim is not well supported. In Table 1, the author should also provide the detailed model size and inference speed. So the readers will better understand how much to pay to obtain such performance.

3. The MoST-Grounding module with both temporal and spatial localizers is effective: This is supported by the ablation study. The study removes the spatial localizer, showing a drop in average accuracy on NExT-QA, indicating the usefulness of spatial grounding. Further,  the evaluation on NExT-GQA demonstrates accurate temporal grounding.

**Essential References Not Discussed:**

I do not have recommendations on essential references.

**Experimental Designs Or Analyses:**

1. The author should explicitly show the full parameter sizes in Table 1. It is not clear how the model sizes are different between different models. The reason why the model size is important is because the proposed method includes multiple off-the-shelf expert models to attend to specific frames both temporally and spatially. Including the model size will help the readers to understand the cost.
2. Further, the experimental results lack the comparison of training and inference efficiency with other methods. With many add-on modules, what is the throughput or the inference speed when comparing with other models?
3. Another concern is about the improvement in Table 1. From my point of view, the improvement is not significant when comparing with the direct baselines. For example, when comparing to LLaVA-Next, the performance increases 1.8% on the NExt-QA benchmark. Consider the increased model size and lower inference speed, how would the author address their advantages?

**Methods And Evaluation Criteria:**

The proposed method uses multiple 'expert' modules to help localize frames both temporally and spatially. Further,  self-referential training is achieved by reinforcement learning on the Question Parser.

**Other Comments Or Suggestions:**

I do not have other comments or suggestions.

**Other Strengths And Weaknesses:**

Please refer to previous comments.

**Questions For Authors:**

I do not have other questions.

**Relation To Broader Scientific Literature:**

This paper is majorly related to the multimodal LLM and video question answering.

**Theoretical Claims:**

There are no significant theoretical claims in this paper. The author attempts to provide the computational complexity analysis in Section 3.5.

However, it is not clear what P1 - P3 mean in Proposition 3.1, and where does P4 come from?

---

> ### Author Rebuttal · Authors · 2025-03-31
>
> We sincerely thank the reviewer for taking time to review our paper and providing insightful feedback and suggestions. We address the weaknesses and questions as follows:
>
> ### **Claims And Evidence**
>
> 1.The Question Parser's prompt is meticulously crafted. It details each module's function and includes carefully-chosen basic examples, covering common and corner cases, to guide policy generation. As seen in Figures 4,5,6, the generated policy is a concise JSON string, not some heavy structure. Although the prompt is long due to multiple policy examples, this is essential for the Question Parser to produce reasonable policies.
>
> 2.Please refer to "Experimental Designs Or Analyses" part.
>
> ### **Theoretical Claims**
>
> We are sorry about the typo in Preposition 3.1. Preposition 3.1 should be:
>
> Given parameters of Multimodal LLM $P_1$, $P_2$, $P_3$ and $P_4$, video with $N$ input frames and resolution $H \times W$, text input with length $l$, the complexity of Multimodal Answerer is $O\left(P_1NH^2W^2 + P_2N^2 + P_3l^2 + P_4Nl \right)$.
>
> As we have only selected Qwen-VL as the base model to analyze the computational complexity, $P_1$, $P_2$, $P_3$ and $P_4$ represent constants relevant to parameters of Qwen-VL. Specifically, according to the proof of Preposition G.5 in Appendix G, given the parameters of the vision transformer $d_{VT}, L_{VT}, d_{ff_{VT}}$, the parameters of QwenLM are $d_Q, L_Q, d_{ff_Q}$, the number of queries in the cross attention layer $n_q$ and the kernel size and stride in the convolution layer $s$, we have: $P_1 = \frac{2L_{VT}d_{VT}}{s^4}$, $P_2 = 2L_Qd_Qn_q^2$, $P_3 = 2L_Qd_Q$, $P_4 = 4L_Qd_Qn_q$.
>
> ### **Experimental Designs Or Analyses**
>
> || Parameter Size | Inference Speed (s / sample) | Acc on NExT-QA |
> | ------------------------ | :------------: | :--------------------------: | :------------: |
> |BLIP-2|4.1B|1.21|69.6|
> |LSTP*|4.3B|1.62|72.1|
> |InstructBLIP|7.9B|1.75|72.5|
> | SeViLa|8.3B|2.79|      73.8      |
> | Qwen-VL|      9.6B      |             1.32             |      71.9      |
> | MSR-ViR$_Q$(1.5B parser) |     11.2B      |             2.35             |      73.1      |
> | MSR-ViR$_Q$(7B parser)   |     16.7B      |             3.10             |      73.6      |
> | LLaVA-Next|      7.1B      |             2.19             |      73.1      |
> | MSR-ViR$_L$(1.5B parser) |       8.7B       |             4.29             |      74.2      |
> | MSR-ViR$_L$(7B parser)   |     14.2B      |             4.96             |      74.9      |
>
> We test the inference speed of our method and Multimodal LLM-based baselines on an NVIDIA A100 GPU. Results are presented in the above table. We omit GCG from the table as its repository lacks inference code. LSTP, designed for high-efficiency inference with optical flow, is marked. For BLIP-2, LSTP, InstructBLIP, SeViLa and Qwen-VL, we sample 4 frames from the video. For LLaVA-Next, we sample 8 frames from the video. The settings of MSR-ViR keep consistent with those in our paper.
>
> Our framework's additional parameters mainly stem from the Question Parser; MoST-Grounding contributes less than 0.1B. For comprehensive comparison, we test total parameters, inference speed, and accuracy using Question Parsers of different sizes (Qwen2-7B and Qwen2-1.5B). With Qwen2-7B, the inference speed of MSR-ViR is about twice that of the direct baseline, consistent with our complexity estimates. This computational overhead is reasonable, as reasoning-based methods typically require more time to answer questions due to the step-by-step reasoning process, like GPT-o1 over GPT-4o, and DeepSeek-R1 over DeepSeek-V3. With Qwen2-1.5B, although accuracy slightly drops, it still outperforms the direct baseline with fewer additional parameters and less computational overhead.
>
> Although MSR-ViR introduces additional parameters and computational overhead, we address its advantages as follows:
>
> - MSR-ViR provides interpretable reasoning path which is refined by self-reflection training, while previous methods fail to do so. As MoST-Grounding utilizes faster small modules, the computational overhead mainly comes from Question Parser that provides interpretability of our framework, and the computational overhead is reasonable.
> - As for the issue of insignificant improvement in Table 1, please refer to Q1 in our rebuttal to reviewer u3Mw. We claim that the impact of grounding for Multimodal LLM on VideoQA datasets is limited, consistent with previous grounding-based method. Besides, we further conduct experiments on long-form VideoQA dataset VideoMME, where grounding and reasoning are more important, to demonstrate the superiority of our framework.
> - Table 3 demonstrates that MSR-ViR achieves significantly higher grounding accuracy and grounded-QA accuracy than previous methods. The IoU is even higher than that of LLoVi and MoReVQA which utilize significantly larger LLMs GPT-4 and Palm-2. This demonstrates that MSR-ViR is capable of more accurately grounding and answering questions.

---

### Official Review · Reviewer_mAE3 · 2025-03-17

**Overall Recommendation:** 4

**Summary:**

The paper introduces a framework MSR-ViR designed to improve interpretability of multimodal LLMs in VideoQA. Unlike traditional end-to-end multimodal LLMs that function as black boxes, MSR-ViR integrates modular networks to provide explicit reasoning paths. MSR-ViR serially combines (1) a question decomposer that divides complex a question into sub-steps (via few-shot prompting of a LLM), (2) a temporal localizer that localizes the relevant temporal segments (via UniVTG model), (3) a spatial localizer that identifies the spatial segments (via YOLO-World model), (4) and a multimodal LLM that is trained to answer questions given the segments. To further improve this pipeline, a Self-Reflection Training was proposed. The LLM is fine-tuned to output better policy / sub-steps, guided by the loss from the multimodal LLM. Finally, the LLM and the multimodal LLM are trained in a cyclical manner. Through experiments on benchmarks (NExT-QA, STAR, EgoSchema, and NExT-GQA), MSR-ViR demonstrates good performance in video understanding and localization accuracy compared to baseline methods.

### Update After Rebuttal
Thanks the authors for their clarification which address most of my concerns. I raise my score to accept and expect that the authors will revise the paper according to the suggestions from reviewers.

**Claims And Evidence:**

Over-claim on the explicit reasoning paths: One major claim of this work is that the model can provide reasoning paths. However, the path is provided by text-only LLM and based on question only (e.g., Figure 3), instead of multi-modal reasoning path. Besides, the experiments are mostly concerned about QA accuracy, leaving textual reasoning paths less critical.

**Essential References Not Discussed:**

See above.

**Experimental Designs Or Analyses:**

Complex pipeline yet less significant performance improvement: The LLM is trained to better fit with video QA and thus output a more reasonable step plan. The performance improvement is not significant (Table 1-3) and mostly from LLM self-reflection training (Table 4). One possibility is that the combined modules (temporal localizer and spatial localizer) accumulate errors. What if we combine it with an end-to-end video LLM with grounding capability?

**Methods And Evaluation Criteria:**

Yes.

**Other Comments Or Suggestions:**

The layout of Figure 2 can be further improved for better reading experience.

**Other Strengths And Weaknesses:**

Strengths:
1.	This work tries to provide explicit reasoning steps for video QA using multimodal LLMs. This direction is promising and worth exploration.
2.	This work proposes a reinforcement learning strategy that helps improve the reasoning plan and thus final QA accuracy. This design is simple and effective.
3.	The paper writing and figures are generally good.

Weaknesses: See above sections.

**Questions For Authors:**

I would consider raising the scores depending on the answers to the concerns / weaknesses above.

**Relation To Broader Scientific Literature:**

Missing references: The overall framework can be considered as using external modules / tools to obtain structured and additional information for supporting multimodal LLM reasoning. It would be great if the authors could discuss the works in this direction, such as [A, B]. Besides, one key improvement of this work comes from the reinforcement learning supervised by the loss of generating answers. Similar key idea was proposed by [C] which tries to improve caption quality via sentence metric.

[A] Beyond Embeddings: The Promise of Visual Table in Visual Reasoning, EMNLP 2024

[B] MM-Reasoner: A Multi-Modal Knowledge-Aware Framework for Knowledge-Based Visual Question Answering, EMNLP 2023

[C] Self-critical Sequence Training for Image Captioning, CVPR 2017

**Theoretical Claims:**

Yes.

---

> ### Author Rebuttal · Authors · 2025-03-31
>
> We sincerely thank the reviewer for taking time to review our paper and providing insightful feedback and suggestions. We address the weaknesses and questions as follows:
>
> ### **Claims And Evidence**
>
> Providing interpretable reasoning paths for black-box Multimodal LLMs in the scenario of VideoQA is one of the major claims in our paper. Although the reasoning path is generated by a text-only LLM, the reasoning process of MSR-ViR is multimodal, as presented in Figure 8, 9, 10. MoST-Grounding localizes the temporal segments and spatial regions related to the question from the video step by step according to the reasoning path. This reasoning process requires not only the textual information in the question but also the visual information in the video to perform step-by-step grounding, and thus it is multimodal.
>
> Due to the lack of label annotations, we are unable to directly evaluate the reasoning path. However, this does not mean the reasoning path is not critical in our framework. Keeping small modules in MoST-Grounding fixed, the reasoning path determines the final spatial-temporal grounding result input to the Multimodal LLM. Therefore, it is crucial for whether the Multimodal LLM can correctly answer the questions and whether the framework can provide accurate grounding results. For this reason, the evaluation of metrics such as QA accuracy, Acc@GQA, IoU is an indirect evaluation of the reasoning path. For example in Table 4, through these metrics, we verify that self-reflection training can improve the quality of the reasoning paths generated by the Question Parser.
>
> To further verify the importance of the reasoning path, we provide another ablation study by using MoST-Grounding to directly provide grounding results with UniVTG and YOLO-World based on the original question without the reasoning path. The results on NExT-GQA are shown in the following table. It can be seen that without the guidance of the reasoning path, both the accuracy in answering questions and the accuracy of the grounding results have significantly decreased. This further illustrates the necessity of the reasoning path.
>
> || Acc@QA | Acc@GQA | mIoU | IoU@0.5 |
> | ---------------------------------------- | ------ | ------- | ---- | ------- |
> | MSR-ViR$_{Q}$|69.9|18.5|22.8|16.4|
> | MSR-ViR$_{Q}$ without self-reflection training |68.3|17.9|22.2|15.7|
> | MSR-ViR$_{Q}$ without reasoning path|66.8|14.4|18.5|11.4|
>
> ### **Experimental Designs Or Analyses**
>
> From the experiment results, it can be seen that the MSR-ViR is capable of providing reasonable reasoning paths, and thereby improves the accuracy of Multimodal LLM in VideoQA. As can be seen in Table 4, self-reflection training plays a key role in improving QA accuracy, which is in line with expectations and is also one of the main contributions of this paper. From the additional ablation study in the above table, it can also be seen that reasoning paths and self-reflection training have significantly improved the performance of grounded-QA, alleviating error accumulation of temporal localizer and spatial localizer. Directly using a video LLM with grounding capability provides a solution to grounding-based VideoQA, but the black-box end-to-end video LLM contradicts the core issue to be addressed in our paper - the issue of interpretability, and thus is not within the scope of discussion in our paper.  For the issue of insignificant performance improvement, please refer to Q1 in our rebuttal to reviewer u3Mw. We claim that the impact of grounding for Multimodal LLM on VideoQA datasets is limited, consistent with previous grounding-based method. Besides, we further conduct experiments on long-form VideoQA dataset VideoMME, where grounding and reasoning are more important, to demonstrate the superiority of our framework.
>
> ### **Relation To Broader Scientific Literature**
>
> We thank the reviewer for the suggestions of missing references, and discuss these works as follows:
>
> [A] introduces Visual Table, a novel visual representation that provides detailed object descriptions and knowledge in structured text, significantly boosting performance in visual reasoning tasks, while [B] presents MM-Reasoner, which leverages vision APIs and LLMs to extract and utilize query-specific knowledge. Other works explore the use of outputs for reinforcement learning of the model, such as [C] that improves image captioning quality by normalizing rewards with test-time inference outputs, leading to significant performance gains. These works share some similarities with certain ideas in our work. However, they are all focused on image tasks, rather than VideoQA task which involves more complex scenarios and requires more grounding and reasoning. We will add detailed discussion to the final version of our paper.
>
> ### **Other Comments Or Suggestions**
>
> We will make revisions to the layout of Figure 2 for clarity and comprehensibility.

---

### Decision · Program_Chairs · 2025-05-01

**Decision:**

Accept (poster)

**Comment:**

The paper presents MSR-ViR, a new method for interpretable video question answering, combining multiple modular networks with a multimodal LLM. It employs an Alternate Self-reflection Training Strategy to enhance reasoning by optimizing policy generation and the multimodal LLM. Experiments across various datasets demonstrate improved video understanding and answer evidence localization accuracy. All reviewers agree to accept the paper.